# PSEUDO- VS. TRUE-RANDOMNESS: RETHINKING DISTORTION-FREE WATERMARKS OF LANGUAGE MODELS UNDER WATERMARK KEY COLLISIONS

## ABSTRACT

Language model (LM) watermarking techniques inject a statistical signal into LM-generated content by substituting the random sampling process with pseudo-random sampling, using watermark keys as the random seed. Among these statistical watermarking approaches, distortion-free watermarks are particularly crucial because they embed watermarks into LM-generated content without compromising generation quality. However, one notable limitation of pseudo-random sampling compared to true-random sampling is that, under the same watermark keys (i.e., *key collision*), the results of pseudo-random sampling exhibit correlations. This limitation could potentially undermine the distortion-free property. Our studies reveal that key collisions are inevitable due to the limited availability of watermark keys, and existing distortion-free watermarks exhibit a significant distribution bias toward the original LM distribution in the presence of key collisions. Moreover, we go beyond the key collision condition and prove that achieving a perfect distortion-free watermark is impossible. To study the trade-off between watermark strength and its distribution bias, we introduce a new family of distortion-free watermarks–beta-watermark. Experimental results support that the beta-watermark can effectively reduce the distribution bias under key collisions.

## 1 INTRODUCTION

In an era where artificial intelligence surpasses human capabilities in generating text, the authenticity and origin of such AI-generated content have become paramount concerns. Language model watermarking (Aaronson, 2022; Kirchenbauer et al., 2023; Christ et al., 2023; Kuditipudi et al., 2023; Hu et al., 2023) provides a promising solution for distinguishing between human and machine-generated text. This technique secretly embeds a statistical signal into the generated text using a pseudo-random generator seeded with watermark keys. The embedded signal is then detected through a statistical hypothesis test, ensuring the traceability and verification of the text's origin.

Distortion-free watermarks (Aaronson, 2022; Christ et al., 2023; Kuditipudi et al., 2023; Hu et al., 2023) represent one of the most compelling techniques in language model watermarking. These watermarks are particularly valuable because they provably preserve the output distribution of the original language model. Specifically, the expected watermarked distribution with respect to the watermark keys remains identical to the original language model distribution, thus offering significant practical application potential.

However, the pseudo-random nature of the watermark generator may lead to correlations between generated content when the watermark keys are identical (i.e., key collision). In extreme cases, such as when the prompt remains the same, key collisions can result in identical generated content, significantly limiting its application scenarios. For instance, when using GPT-4 to generate content, if the initial output is unsatisfactory, a request to regenerate would typically yield a different result. However, under a distortion-free watermarking scheme, the output may remain unchanged due to the consistent application of the same watermark key. This limitation highlights a critical challenge in the practical deployment of such watermarking techniques.

In our research, we comprehensively analyze the existing distortion-free watermarks and demonstrate, through both theoretical and empirical evidence, that *no distortion-free watermark can fully preserve*

*the original LM distribution under key collisions.* Specifically, we categorize the level of distortion-free capability into three types: a) Step-wise distortion-free—the watermark preserves the LM distribution at a single token generation step; b) Weakly distortion-free—the watermark preserves the LM distribution for a one-time sentence generation; c) Strongly distortion-free—the watermark preserves the LM distribution across multiple sentence generations. Our findings indicate that all existing distortion-free watermarks are weakly distortion-free but not strongly distortion-free due to key collisions. Under the key collisions, In particular, we theoretically prove that there does not exist any detectable strongly distortion-free watermark. We also show that key collisions are inevitable given the limited number of watermark keys available in current schemes.

To mitigate the distribution bias caused by key collisions, we introduce the beta-watermark and develop a model-agnostic detector that can identify watermarks without requiring access to prompts or language models. Additionally, we design empirical metrics to measure the distribution bias resulting from key collisions. Through rigorous testing on widely-studied language models, including BART-large model (Liu et al., 2020) and LLaMA-2 (Touvron et al., 2023), our beta-watermark has demonstrated effectiveness in significantly reducing the distribution bias induced by key collisions.

Our main contributions are summarized as follows:

- We identify three levels of distortion-free capabilities in watermarks—Step-wise, Weakly, and Strongly Distortion-free—revealing that existing watermarks are not strongly distortion-free and cannot preserve the original language model distribution under multiple generations due to the inevitability of key collisions.

- Under watermark key collisions, we theoretically demonstrate a trade-off between watermark strength and its distribution bias to the original LM distribution—a smaller distribution bias results in weaker watermark strength. Based on our discussion on the distribution bias, we proposed a black-box distortion-free watermark detection approach, which can effectively check if an LM is watermarked given the original LM. Furthermore, we go beyond the key collision assumption and prove that strongly distortion-free watermarks does not exist.

- We introduce beta-watermark, a new family of weakly distortion-free watermarks that can provably reduce the distribution bias by trading the watermarking strength. Through experiments on popular language models like BART-large and LLaMA-2, we demonstrate our theoretical findings that existing watermarks are not strongly distortion-free and beta-watermark can effectively reduce the distribution bias.

## 2 RELATED WORK

**Statistical watermarks.** Kirchenbauer et al. (2023) enhanced the statistical watermark framework originally introduced by Aaronson (2022), demonstrating the effectiveness of statistical watermarking through extensive experiments on large language models. They splited the LM tokens into red and green list, then promoted the use of green tokens by adding a fixed parameter $\delta$ to their logits. Zhao et al. (2023) proposed the unigram watermark, which enhances the robustness of the statistical watermark by using one-gram hashing to produce watermark keys. Liu et al. (2023b) also improved the robustness of statistical watermarking by leveraging the semantics of generated content as watermark keys. Additionally, Liu et al. (2023a) proposed an unforgeable watermark scheme that employs neural networks to modify token distributions instead of using traditional watermark keys. However, these approaches may lead to significant changes in the distribution of generated text, potentially compromising content quality.

**Distortion-free watermarks.** To preserve the original output distribution in watermarked content, researchers have explored alternative strategies to modify the token distribution. Aaronson (2022) introduced the first distortion-free watermarking strategy, which utilized Gumbel-reparametrization to alter token distribution and the prefix n-gram content as the watermark keys. Christ et al. (2023) and Kuditipudi et al. (2023) adopted the inverse-sampling and Gumbel-reparametrization to modify the watermarked token distributions, where the watermark keys are based on the token position or a fixed key list respectively. Notice Christ et al. (2023)'s method encounters resilience challenges under modifications and lacks empirical evidence regarding its detectability. Meanwhile, Kuditipudi et al. (2023)'s detection process involves hundreds of resampling steps from the secret key distribution, proving inefficient for processing lengthy texts. Hu et al. (2023) employed inverse-sampling and

permute-reweight methods for watermarking. But their detector is not model-agnostic, which requires access to the language model API and prompts. Wu et al. (2023) improved the permute-reweight methods and designed a model-agnostic detector. A detailed related work section is in Appendix B.

## 3 PRELIMINARY

**Notations.** Denote by $V := \{t_1, ..., t_N\}$ the vocabulary (or token) set of a language model, and by $N = |V|$ its size. Let $\mathcal{V}$ represent the set of all conceivable string sequences, including those of zero length. A language model generates a token sequence based on a predetermined prompt. For a single step in this process, the probability of generating the next token $x_{n+1} \in V$, given the current context from $x_1$ to $x_n$, is represented as $P_M(x_{n+1} \mid x_1, x_2, \ldots, x_n)$. For brevity, we adopt the condensed notation: $P_M(\boldsymbol{x}_{n+1:n+m} \mid \boldsymbol{x}_{1:n})$, where $\boldsymbol{x}_{n+1:n+m} = (x_{n+1}, \ldots, x_{n+m})$. Note that the prompt is deliberately omitted in this representation. Inherent to its design, the language model operates in an autoregressive mode. This implies that the combined probability of generating several tokens, specifically from $x_{n+1}$ to $x_{n+m}$, takes the form $P_M(\boldsymbol{x}_{n+1:n+m} \mid \boldsymbol{x}_{1:n}) = \prod_{i=1}^{m} P_M(x_{n+i} \mid \boldsymbol{x}_{1:n+i-1})$.

**Watermarking problem definition.** A language model (LM) service provider aims to watermark the generated content such that all other users can verify if the content is generated by the LM without needing access to the LM or the original prompt. A watermark framework primarily consists of two components: a *watermark generator* and a *watermark detector*. The watermark generator embeds a watermark into the text through a *Pseudo-random Distribution Adjustment rule* (PDA-rule), which is seeded by watermark keys. The watermark detector, on the other hand, detects the presence of the watermark within the content using a statistical hypothesis test.

**Definition 3.1** (PDA-rule). *Let $\mathcal{P}$ represent the space of token distributions and let $K$ denote the space of watermark keys. A Pseudo-random Distribution Adjustment rule (**PDA-rule**), defined as $F : \mathcal{P} \times K \to \mathcal{P}$, adjusts the token distribution based on a given watermark key.*

**Watermark generator.** During the watermark generation process, the service provider modifies the original language model distribution $P_M$ using a *watermark key* $k \in K$ and a PDA-rule. Here, the watermark key acts as a random seed to modify the distribution, after which the next token is sampled from this modified distribution. A watermark key usually consists of a *secret key* sk and a context key (e.g., n-gram (Aaronson, 2022) or token position (Christ et al., 2023)). Let $\mathcal{F} := \{F : \mathcal{P} \times K \to \mathcal{P}\}$ denote the set of PDA-rules. Specifically, let $P_W$ denote the distribution of the LM after watermarking, and $k$ the watermark key, $P_W(t \mid \boldsymbol{x}_{1:n-1}) := F(P_M(\cdot \mid \boldsymbol{x}_{1:n-1}), k)(t), \forall t \in V$, where $P_M(\cdot \mid \boldsymbol{x}_{1:n-1})$ is the LM token distribution for sampling the $n$-th token. When sampling the next token $x_n$, the language model samples from $P_W(\cdot \mid \boldsymbol{x}_{1:n-1})$ instead of $P_M(\cdot \mid \boldsymbol{x}_{1:n-1})$. This mechanism allows the service provider to inject a statistical signal into the generated content.

The PDA-rule is the core of the watermark generator. A PDA-rule is considered distortion-free if and only if it preserves the token distribution during watermark generation. To the best of our knowledge, there are three types of distortion-free PDA-rules: inverse-sampling (Christ et al., 2023; Kuditipudi et al., 2023; Hu et al., 2023), Gumbel-reparametrization (Aaronson, 2022; Kuditipudi et al., 2023; Fu et al., 2024), and permute-reweight (Hu et al., 2023). A detailed introduction to these methods can be found in Section 4.1. The formal definition of a distortion-free PDA-rule is presented below.

**Definition 3.2** (Distortion-free PDA-rule). *A PDA-rule $F$, is a distortion-free PDA-rule, if and only if for an arbitrary LM $P_M$, $\forall \boldsymbol{x}_{1:n} \in \mathcal{V}$, and $\forall i \leq n$, it holds that $P_M(x_i|\boldsymbol{x}_{1:i-1}) = \mathbb{E}_{k_i}[F(P_M(\cdot|\boldsymbol{x}_{1:i-1}), k_i)(x_i)]$.*

**Watermark Detector.** During the process of watermark detection, the user will have access only to the watermark key and the PDA-rule $F$. The detector employs a hypothesis testing approach to identify the presence of the watermark signal. The hypothesis test is defined as: $H_0$ : *The content is generated without the presence of watermarks*, and $H_1$ : *The content is generated with the presence of watermarks*. For the purposes of the statistical test, a score function $s(x, k, F) : V \times K \times \mathcal{F} \to \mathbb{R}$ is employed. Under $H_0$, the score function is a random variable $S_{H_0}$ where $\Pr(S_{H_0} = s(t, k, F)|k, F) = \sum_{s(t', k, F) = s(t, k, F)} P_M(t'), \forall t \in V$, while under $H_1$, the random variable $S_{H_1}$ becomes $\Pr(S_{H_1} = s(t, k, F)|k, F) = \sum_{s(t', k, F) = s(t, k, F)} P_W(t')$. Thus, we can use the discrepancy between $S_{H_0}$ and $S_{H_1}$ to detect the watermark content. Given an observation (text sequence) $\boldsymbol{x}_{1:n}$, we define the test statistic $S(\boldsymbol{x}_{1:n}) = \sum_{i=1}^{n} s(x_i, k, F)$ as the measure for the test.

The decision to reject the null hypothesis is based on the difference between $S(\boldsymbol{x}_{1:n})$ and the expected value $\mathbb{E}_{H_0}[S(\boldsymbol{x}_{1:n})]$.

**Watermark Key.** For each generating step, we will use a watermark key to seed the PDA-rule. There are generally three key sampling methods:

- **(n-gram hashing)** Aaronson (2022), Christ et al. (2023) and Hu et al. (2023) use a fixed **secret key** $\mathsf{sk}_0$ and the prefix n-gram $s$ (e.g., $s = \mathbf{x}_{l-n:l-1}$ for generating $x_l$) to form the watermark keys, i.e., $K = \{(\mathsf{sk}_0, s) \mid s \in \mathcal{V}_n\}$, where $\mathcal{V}_n$ represents the set of all n-grams with token set $V$. A history list is kept during one generation to ensure the watermark keys are unique. If the length of previously generated tokens is less than $n$, all preceding tokens are used as $s$.

- **(position hashing)** Christ et al. (2023) uses a fixed **secret key** $\mathsf{sk}_0$ and the token position are used as watermark keys, i.e., $K = \{(\mathsf{sk}_0, i) \mid i \in \mathbb{N}\}$.

- **(fixed key set)** Kuditipudi et al. (2023) uses a fixed **secret key** $\mathsf{sk}_0$ generates a set of watermark keys, $K = \{k_1, \ldots, k_{n_0}\}$. During token generation at step $i$, a random integer $r$ is sampled, and $k_{(i+r) \mod n_0}$ is used as the seed for the PDA-rule. If the token length exceeds $n_0$, we will sample from the original LM distribution instead.

**Definition 3.3** (Key collision). *Key collision refers to scenarios where the same watermark keys are used to seed the PDA-rule.*

All three watermark key sampling methods mentioned previously have a limited number of keys given the fixed secret key $\mathsf{sk}_0$. The maximum key volume is $|V|^n$ for n-gram hashing, $l_0$ for position hashing, and $n_0$ for the fixed key set. Here, $l_0$ represents the maximum token length for the language model, typically ranging from $10^4$ to $10^6$. Therefore, if we only have one secret key, key collisions will occur when the number of queries and the generated tokens exceeds the key volume.

## 4 CURSE OF KEY COLLISION ON DISTORTION-FREE WATERMARKS

We start with showing key collision is inevitable. In the previous section, we show that given a fixed secret key $\mathsf{sk}_0$, the watermark key space is finite. Consequently, key collisions will occur with a sufficient number of queries to the language model. One might naturally question whether using an infinite number of secret keys (e.g., a unique key for each generation) could expand the watermark key space to infinity, thereby reducing the likelihood of collisions. However, this approach is impractical because it would substantially reduce detection efficiency. When analyzing a watermarked sequence, the detection algorithm would need to be applied to all possible secret keys, even though only one key corresponds to the watermark. Thus, the watermark information becomes obscured by the numerous other keys. All missing proofs can be found in Appendix D.

**Lemma 4.1** (Detection efficiency with multiple secret keys). *Denote by $S(\cdot | \mathsf{sk})$ the test statistic. Under the null hypothesis $H_0$, given a random text $\boldsymbol{x}_{1:n}$, we have $\Pr(S(\boldsymbol{x}_{1:n} | \mathsf{sk}_0) - \mathbb{E}_{H_0}[S] \geq t | H_0) = p_0(t)$, i.e., $p_0(t)$ is the false positive rate of threshold $t$ under single secret key detection. Given $M$ different secret keys, if we use the maximum of the score as the test statistic, we have*

$$\Pr\left(\max_{i \in [M]}(S(\boldsymbol{x}_{1:n} | \mathsf{sk}_i) - \mathbb{E}_{H_0}[S]) \geq t | H_0\right) = 1 - (1 - p_0(t))^M, \quad \forall t \in \mathbb{R}.$$

**Corollary 4.2.** *Under the existing watermark key sampling schemes, key collision is inevitable.*

Lemma 4.1 states that, given the same threshold $t$, the false positive rate increases with the number of secret keys. Especially, when $M \to \infty$, the false positive rate will tend to 1, which indicates every sentence will be detected as watermarked. Thus, the number of secret keys should be finite, and key collision is inevitable.

We then provide the definition of the three levels of distortion-free capabilities in watermarks: 1) distortion-free within a single token generation, 2) distortion-free in one entire generation, 3) distortion-free across multiple generations.

**Definition 4.3** (Step-wise distortion-free watermark). *If a watermark framework adopts a distortion-free PDA-rule, then it is a step-wise distortion-free watermark.*

**Definition 4.4** (Weakly distortion-free watermark). *A step-wise distortion-free watermark $P_W$ is weakly distortion-free, if $\forall n \in \mathbb{N}_+, \forall \boldsymbol{x}_{1:n} \in \mathcal{V}$, we have $P_M(\boldsymbol{x}_{1:n}) = \mathbb{E}_{\boldsymbol{k}_{1:n}}[P_W(\boldsymbol{x}_{1:n}|\boldsymbol{k}_{1:n})]$.*

**Definition 4.5** (Strongly distortion-free watermark). *A step-wise distortion-free watermark $P_W$ is strongly distortion-free if for arbitrary number of generation $N_0$ and $\forall \boldsymbol{x}_{1:n}^{(i)} \in \mathcal{V}, i \in [N_0]$, it holds that $\prod_{i=1}^{N_0} P_M(\boldsymbol{x}_{1:n}^{(i)}) = \mathbb{E}_{\boldsymbol{k}_{1:n}^{(1)}, \dots, \boldsymbol{k}_{1:n}^{(N_0)}}[\prod_{i=1}^{N_0} P_W(\boldsymbol{x}_{1:n}^{(i)}|\boldsymbol{k}_{1:n}^{(i)})]$.*

In the next theorem, we show the sufficient conditions for achieving a weakly/strongly distortion-free watermark.

**Theorem 4.6.** *A watermark framework is a weakly/strongly distortion-free watermark if a) it adopts a distortion-free PDA-rule and b) there is no key collision during watermark generation.*

**Corollary 4.7.** *A watermark that consists of a distortion-free PDA-rule with n-gram hashing, position hashing or fixed key set is a weakly distortion-free watermark.*

The proof of this corollary is straightforward because all these watermark key samplers guarantee the uniqueness of each watermark key in a single generation. However, across multiple generations, key collisions become inevitable as the number of generated tokens can surpass the volume of available keys. In the rest of this section, we will explain how key collisions can impact the generation quality and lead to a biased watermarked distribution compared to the original language model distribution.

## 4.1 EXISTING DISTORTION-FREE PDA-RULES

To analyze the influence of key collision on the distortion-free watermarks, we begin with introducing the existing PDA-rules. We also provide a detailed illustration of the existing PDA-rules in Figure 1.

**Gumbel-reparametrization.** In the Gumbel-reparametrization rule, when sampling $x_i$ with the watermark key $k_i$, we first sample Gumbel pseudo-random variables $g_1(k_i), \dots, g_N(k_i) \sim Gumbel(0,1)$ with the watermark key $k_i$. These $N$ independent Gumbel random variables are added to the log-probability of tokens $\log P_M(t_1|\boldsymbol{x}_{1:i-1}), \dots, \log P_M(t_N|\boldsymbol{x}_{1:i-1})$. The token that achieves the maximum value is then selected as the next token $x_i$. This process can be formulated through the following equation: $F_{GR}(P_M(\cdot|\boldsymbol{x}_{1:i-1}), k_i) = \delta_{t_{m^*}}$, where $m^* = arg\max_{m \in [N]}(g_m(k_i) + \log P_M(t_m|\boldsymbol{x}_{1:i-1}))$ and $\delta$ is the Dirac function.

**Inverse-sampling.** In the inverse-sampling rule, when sampling $x_i$ with the watermark key $k_i$, we first organize the LM token probability $P_M(t_1|\boldsymbol{x}_{1:i-1}), \dots, P_M(t_N|\boldsymbol{x}_{1:i-1})$ within the interval $[0,1]$. Then we will sample a pseudo-random variable $r(k_i) \in U(0,1)$, where $U(0,1)$ is the uniform distribution on $[0,1]$. The next token is selected based on the location of $r(k_i)$ within the cumulative probability intervals on $[0,1]$. This process can be formulated through the following equation: $F_{IS}(P_M(\cdot|\boldsymbol{x}_{1:i-1}), k_i) = \delta_{t_{m^*} \in V}$, where $r(k_i) \in [\sum_{j=1}^{m^*-1} P_M(t_j|\boldsymbol{x}_{1:i-1}), \sum_{j=1}^{m^*} P_M(t_j|\boldsymbol{x}_{1:i-1})]$ and $\delta$ is the Dirac function.

**Permute-reweight.** In the permute-reweight rule, when sampling $x_i$ with the watermark key $k_i$, we first generate a pseudo-random token permutation $\pi(\cdot|k_i) : V \to [N]$, which is a bijection between token set $V$ and $[N]$. The token permutations are uniformly distributed with the watermark keys. The LM token probabilities are then rearranged within the interval $[0,1]$ according to the permutation $\pi(\cdot|k_i)$. The token probability within $[0, 1/2]$ will be scaled to 0, and the rest half will be scaled to 1. Subsequently, $x_i$ is randomly sampled following this adjusted distribution. We can formulate the permute-reweight rule through the following formula: $F_{PR}(P_M(\cdot|\boldsymbol{x}_{1:i-1}), k_i)(t) = \max\{2\sum_{t', \pi(t'|k_i) \leq \pi(t|k_i)} P_M(t'|\boldsymbol{x}_{1:i-1}) - 1, 0\} - \max\{2\sum_{t', \pi(t'|k_i) \leq \pi(t|k_i)-1} P_M(t'|\boldsymbol{x}_{1:i-1}) - 1, 0\}$.

**Pseudo- vs True- Randomness.** Based on the above discussion, it is clear that token sampling using Gumbel-reparametrization or inverse-sampling relies entirely on pseudo-randomness, as the watermark distribution for these methods is deterministic given the watermark key. Consequently, for the same token distribution, key collisions result in identical token generation. For instance, when **generating multiple responses with the same prompt**, the first token will always be identical. In contrast, token sampling with the permute-reweight rule does not fully depend on pseudo-randomness. The permute-reweight PDA-rule only scales the first half of the distribution to zero, preserving the rest of the token probabilities. True-random sampling is then applied to the remaining tokens.

Table 1: Summarization of existing distortion-free watermarks.

| | | Aaronson (2022) | Christ et al. (2023) | Kuditipudi et al. (2023) | Hu et al. (2023) |
|---|---|---|---|---|---|
| Watermark generator | PDA-rule | Gumbel-reparametrization | Inverse-sampling | Inverse-sampling, Gumbel-reparametrization | Inverse-sampling, Permute-reweight |
| | Watermark key sampler | n-gram hashing | position hashing | fixed key set | n-gram hashing |
| Watermark detector | Model-agnostic | ✓ | ✓ | ✓ | ✗ |
| | Robust | ✓ | ✗ | ✓ | ✓ |
| Level of distortion-free | Step-wise distortion-free | ✓ | ✓ | ✓ | ✓ |
| | Weakly distortion-free | ✓ | ✓ | ✓ | ✓ |
| | Strongly distortion-free | ✗ | ✗ | ✗ | ✗ |

Figure 1: Pseudo-randomness in a token sampling step for watermarked LMs. "Before" refers the original LM token distribution and "After" refers the watermarked token distribution. Given a fixed watermark key, both inverse-sampling and Gumbel reparametrization methods become deterministic. In contrast, the permute-reweight method retains elements of randomness.

## 4.2 DISTRIBUTION BIAS OF DISTORTION-FREE WATERMARKS UNDER KEY COLLISIONS

In this subsection, we explore the distribution bias introduced by the watermark. Given that the distribution overlap between two distributions $P_1, P_2 \in \mathcal{P}$ is represented by $\sum_{t \in V} \min\{P_1(t), P_2(t)\}$, we use $1 - \sum_{t \in V} \min\{P_1(t), P_2(t)\}$, i.e., the total variation, to measure the distribution bias between $P_1$ and $P_2$. Under the key collisions, the bias introduced by a PDA-rule $F$ on a token distribution $P \in \mathcal{P}$ is $1 - \sum_{t \in V} \min\{P(t), F(P|k)(t)\}$. Thus, we introduce the *expected total variation* as a metric for measuring distribution bias.

**Definition 4.8** (Expected total variation). *Given a token distributions $P \in \mathcal{P}$ and a PDA-rule $F$, the expected total variation between them is given by $\mathbb{D}(P, F) := 1 - \mathbb{E}_k[\sum_{t \in V} \min\{P(t), F(P|k)(t)\}]$.*

**Trade-off between watermark strength and distribution bias under key collisions.** Interestingly, the expected total variation also reflects the watermark's strength. In statistical watermarking, where the goal is to embed a statistical signal into generated content, a larger total variation enhances the strength of this signal and improve the detection efficiency. However, under key collisions, it is desirable for the expected total variation to be minimized to better preserve the original LM distribution. Therefore, a trade-off exists between watermark strength and distribution bias under key collisions.

We compute the expected distribution bias of the existing distortion-free PDA-rules: Gumbel-reparametrization $F_{GR}$, inverse-sampling $F_{IS}$, and permute-reweight $F_{PR}$.

**Theorem 4.9.** *Given an arbitrary token distribution $P \in \mathcal{P}$, we have*

$$\mathbb{D}(P, F_{GR}) = \mathbb{D}(P, F_{IS}) = 1 - \sum_{t \in V} P(t)^2,$$

*and*

$$0.5(1 - \max_{t \in V} P(t)) \leq \mathbb{D}(P, F_{PR}) \leq 0.5 - \max\{\max_{t \in V} P(t) - 0.5, 0\}.$$

*Moreover, $\mathbb{D}(P, F_{PR}) \leq \mathbb{D}(P, F_{IS}) = \mathbb{D}(P, F_{GR})$.*

From this theorem, we find that the permute-reweight watermark exhibits a smaller distribution bias compared to the Gumbel-reparametrization and inverse-sampling watermarks. This finding aligns with our analysis in Section 4.1, where we assert that Gumbel-reparametrization and inverse-sampling become deterministic with a fixed watermark key, while permute-reweight maintains an element of randomness, resulting in a smaller distribution bias. In the next theorem, we will show

that under key collisions, a watermark with a PDA-rule $F$ is strongly distortion-free if and only if $\mathbb{D}(P, F) = 0, \forall P \in \mathcal{P}$, which indicates that no signal can be embedded into the generated content.

**Theorem 4.10.** *Under key collisions, a watermark with a distortion-free PDA-rule $F$ is strongly distortion-free if and only if $\forall P \in \mathcal{P}$, $\mathbb{D}(P, F) = 0$.*

By integrating Theorem 4.10 with Theorem 4.9, we find that $F_{GR}$, $F_{IS}$, and $F_{PR}$ are unable to yield a strongly distortion-free watermark when key collisions occur. Thus, all existing distortion-free watermarks (Aaronson, 2022; Christ et al., 2023; Kuditipudi et al., 2023; Hu et al., 2023) are not strongly distortion-free. Following the above discussion, we summarize the characteristics of existing distortion-free watermarks in Table 1.

**Corollary 4.11.** *Under key collisions, a strongly distortion-free watermark does not exist.*

If $\forall P \in \mathcal{P}, \mathbb{D}(P, F) = 0$, the watermarked LM shows no distribution bias towards the original LM under the watermark key, i.e., $\forall k \in K, F(P|k) = P$. In this case, no watermark is added to the generated content. As key collision is inevitable, we can conclude that with the current watermark key sampling approaches, a strongly distortion-free watermark does not exist.

### 4.3 A BLACK-BOX DISTORTION-FREE WATERMARK DETECTION APPROACH

As all watermarking approaches present distribution bias towards the original LM under key collisions, we can naturally design a watermark detection approach for the distortion-free watermarks based on the performance difference between the watermarked and the original LMs.

We define a new metric $\Delta$, which measures the performance gap between the watermarked model and the original LM. For $n$ random prompts $p_1, ..., p_n$ with $m$ responses for each $g_1^{p_i}, ..., g_m^{p_i}$, denoted by Met an arbitrary performance metric (e.g., perplexity), $P_M$ the original LM, $P_T$ the test LM, we define $\Delta\text{Met}(P_M, P_T) = \frac{1}{n} \sum_{i=1}^{n} \frac{1}{m} |\sum_{j=1}^{m} \text{Met}(g_j^{p_i}(P_M)) - \sum_{j=1}^{m} \text{Met}(g_j^{p_i}(P_T))|$. Our watermark detection statistic is given by $\text{DetWmk}(P_M, P_T) := \Delta\text{Met}(P_M, P_T) - \Delta\text{Met}(P_M, P_{M'})$, where $P_{M'}$ is identically distributed with $P_M$

**Theorem 4.12.** *If $P_T$ is identically distributed with $P_M$, and $\forall g, |Met(g)| \leq A$, we have $\forall t > 0$,*

$$\Pr(|\textbf{DetWmk}(P_M, P_T)| \geq t) \leq \exp(-\frac{m^2 n t^2}{12A^4}) \tag{1}$$

With this concentration bound, we can efficiently detect whether a language model has been watermarked by examining the performance gap between the test model and the original model. Theorem 4.12 provides a statistical guarantee that if the test model $P_T$ is identically distributed with the original model $P_M$ (i.e., unwatermarked), the probability that our detection statistic $\text{DetWmk}(P_M, P_T)$ exceeds a threshold $t$ diminishes exponentially with the number of prompts $n$ and responses $m$. Specifically, the bound ensures that false positives are highly unlikely when the performance metric is bounded by $A$. This allows us to confidently and efficiently identify watermarked language models by detecting significant deviations in performance metrics.

### 4.4 BEYOND KEY COLLISION - STRONGLY DISTORTION-FREE WATERMARK DOES NOT EXIST

We extend our analysis on strongly distortion-free watermarks and prove that a detectable, strongly distortion-free watermark does not exist. From the above analysis we know that the independence of PDA-rule is a necessary condition for strongly distortion-free watermark, and the independence of PDA-rule stems from the independence of hashed watermark keys $h(k)$. Thus, we can divide the proof into two parts: 1) A strongly distortion-free watermark must use independent hashed watermark keys, denoted as $h(k)$, where $h$ is the hash function employed by the PDA-rule. 2) A watermark using a distortion-free and independent PDA-rule is undetectable by arbitrary detector. Combining 1) and 2) we have the following theorem:

**Theorem 4.13.** *A detectable strongly distortion-free watermark does not exist.*

Theorem 4.13 establishes a fundamental limitation in the design of watermarking schemes by stating that a detectable, strongly distortion-free watermark cannot exist. This theorem also highlights the trade-off in watermarking systems between distribution bias and watermark strength (see Sec. 4.2). If

a watermark is designed to be unbiased to the original data distribution (strongly distortion-free), it cannot be reliably detected using standard detection methods. Conversely, introducing detectability requires some form of alteration or pattern that can be recognized, which compromises the strongly distortion-free property.

# 5 REDUCING DISTRIBUTION BIAS VIA BETA-WATERMARK

In this section, we introduce a new family of watermarks, beta-watermark, which trades watermark strength for low distribution bias. The beta-watermark is based on a distortion-free beta PDA-rule and n-gram hashing. Additionally, we present a model-agnostic detection method for it. In Appendix A Alg. 1 and 2 we show the algorithms of the generator and detector of beta-watermark.

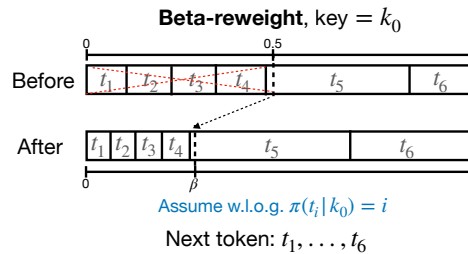

Figure 2: Illustration of Beta PDA-rule.

The beta PDA-rule is a variation of the permute-reweight PDA-rule (another example is DiPmark (Wu et al., 2023)) that introduces greater true randomness during sampling. Similar to permute-reweight watermark, When sampling $x_i$ with the watermark key $k_i$, we first generate a pseudo-random token permutation $\pi(\cdot|k_i) : V \to [N]$. Then we arrange the LM token probability within the interval $[0, 1]$ following the permutation $\pi(\cdot|k_i)$. The first half of token probability (token probability within $[0, 1/2]$) will be scaled to $\beta$, and the rest half probability will be scaled to $1 - \beta$ (See Figure 2 for a detailed illustration). The next token is randomly sampled from the new distribution. Notice, when $\beta = 0$, the permute-reweight PDA-rule is applied and when $\beta = 0.5$, the original LM distribution is used.

**Definition 5.1** (Beta PDA-rule). *Beta PDA-rule $F_\beta$ is defined by:* $F_\beta(P_M(\cdot|\boldsymbol{x}_{1:i-1}), k_i)(t) = (1 - \beta)F_{PR}(P_M(\cdot|\boldsymbol{x}_{1:i-1}), k_i)(t) + \beta[\max\{2\sum_{t', \pi(t'|k_i) \geq \pi(t|k_i)} P_M(t'|\boldsymbol{x}_{1:i-1}) - 1, 0\} - \max\{2\sum_{t', \pi(t'|k_i) \geq \pi(t|k_i)+1} P_M(t'|\boldsymbol{x}_{1:i-1}) - 1, 0\}]$. Notice, the range of $\beta$ is from 0 to 0.5.*

**Theorem 5.2.** *Beta PDA-rule is a distortion-free PDA-rule, i.e., $\forall \boldsymbol{x}_{1:n} \in \mathcal{V}, \forall i \leq n$, $P_M(x_i|\boldsymbol{x}_{1:i-1}) = \mathbb{E}_{k_i}[F(P_M(\cdot|\boldsymbol{x}_{1:i-1}), k_i)(x_i)]$.*

**Corollary 5.3.** *Beta-watermark is a weakly distortion-free watermark.*

The proof is straightforward, as the beta-watermark consists of a distortion-free PDA-rule and the n-gram hashing. In the subsequent theorem, we theoretically demonstrate that the beta PDA-rule introduces a smaller distribution bias compared to the permute-reweight watermark.

**Theorem 5.4.** *Given an arbitrary token distribution $P \in \mathcal{P}$, $\mathbb{D}(P, F_\beta) \leq \mathbb{D}(P, F_{PR}) - \beta(1 - \max_{t \in V} P(t))$. Besides, if $\beta_1 < \beta_2$, $\mathbb{D}(P, F_{\beta_1}) > \mathbb{D}(P, F_{\beta_2})$.*

As the detector of the permute-reweight watermark (Hu et al., 2023) is dependent on the logits from the original LM, we design a new model-agnostic detection algorithm for the beta-watermark. As shown in Figure 2, beta-reweighting tends to enhance the token probability towards the end of the permutation. During detection, given an input token, we can determine its position within the permutation using $\pi(x|k)$. Thus, a higher score should be assigned to larger values of $\pi(x|k)$. We use a sigmoid function: $\text{sigmoid}(C(\pi(x|k)/|V| - 0.5))$, where $C$ is a scaling parameter, to appropriately scale the scores.

**Definition 5.5** (Model-agnostic beta-reweight detection). *We use score function $s(x, k, F) = sigmoid(C(\pi(x|k)/|V| - 0.5))$ to conduct detection. Given a random observation $\boldsymbol{x}_{1:n}$, under the null hypothesis, we have $\Pr(S(\boldsymbol{x}_{1:n}) - \mathbb{E}_{H_0}[S(\boldsymbol{x}_{1:n})] > t\sqrt{n}|H_0) \leq \exp(-2t^2)$.*

# 6 EXPERIMENTS

Our experimental section consists of two parts. In the first part, we compare the weakly and strongly distortion-free nature of the beta watermark with that of existing watermarks, and validate the trade-off between the watermark strength and distribution bias. In the second part, we evaluate

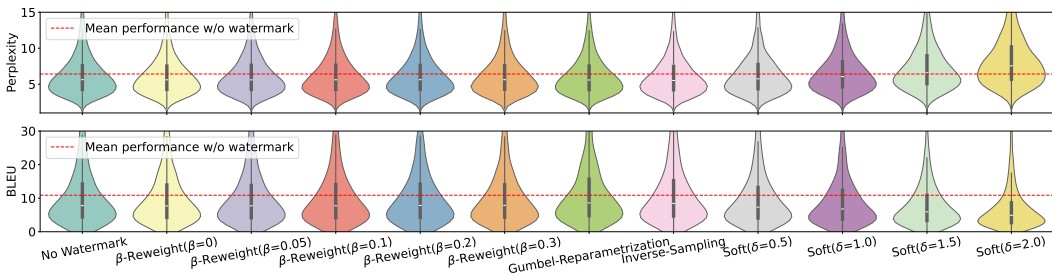

Figure 3: Performance of different watermarks under one-time generation. **Top:** Violin plot of Text Summarization Perplexity. **Bottom:** Violin plot of Machine Translation BLEU. We can see the weakly distortion-free watermarks preserve the generation quality.

Table 2: Performance of different watermarks under multi-time generations. We randomly selected 1000 prompts and generated 100 responses for each. The baseline is the $\Delta$ metrics between two no-watermarked models.

| | Text Summarization | | | Machine Translation | |
|---|---|---|---|---|---|
| | $\Delta$ BERT Score $\downarrow$ | $\Delta$ ROUGE-1$\downarrow$ | $\Delta$ Perplexity $\downarrow$ | $\Delta$ BERT Score $\downarrow$ | $\Delta$ BLEU $\downarrow$ |
| Baseline | 0.0062 | 0.0070 | 0.3028 | 0.0180 | 0.7716 |
| Beta-Reweight ($\beta = 0$) | 0.0090 | 0.0093 | 0.3753 | 0.0267 | 1.2373 |
| Beta-Reweight ($\beta = 0.05$) | 0.0084 | 0.0085 | 0.3549 | 0.0248 | 1.1806 |
| Beta-Reweight ($\beta = 0.1$) | 0.0079 | 0.0081 | 0.3453 | 0.0230 | 1.0316 |
| Beta-Reweight ($\beta = 0.2$) | 0.0070 | 0.0077 | 0.3368 | 0.0203 | 0.9475 |
| Beta-Reweight ($\beta = 0.3$) | **0.0066** | **0.0073** | **0.3144** | **0.0195** | **0.8638** |
| Inverse-sampling | 0.0446 | 0.0494 | 1.7846 | 0.1316 | 5.5354 |
| Gumbel-reparametrization | 0.0428 | 0.0488 | 1.8892 | 0.1341 | 5.6438 |
| Soft($\delta = 1.0$) | 0.0091 | 0.0099 | 0.5473 | 0.0428 | 1.4660 |
| Soft($\delta = 1.5$) | 0.0128 | 0.0136 | 1.1237 | 0.0808 | 2.5310 |
| Soft($\delta = 2.0$) | 0.0195 | 0.0194 | 2.0817 | 0.1274 | 3.7758 |

the detection efficiency of the beta watermark against existing watermarks. In the third part, we assess the robustness of the beta watermark when subjected to random paraphrasing attacks. We focus on three seq2seq tasks in our experiments: machine translation, text summarization and text generation. Detailed experimental settings are provided in Appendix E and additional experimental results, including the detectability and the robustness of beta-watermark, are in Appendix F.

## 6.1 WEAKLY AND STRONGLY DISTORTION-FREENESS

In this section, we conduct experiments to validate our theoretical analysis. We evaluate the weakly and strongly distortion-free properties of existing watermark strategies as defined in Definitions 4.4 and 4.5. We validate the weakly distortion-free property by assessing the quality of the watermarked text generated once for each prompt. For the strongly distortion-free property, we examine the quality of the watermarked text for 1000 prompts, where for each prompt we have 100 generations. We define a new metric $\Delta$, which measures the performance gap between the watermarked model and the original LM. For $n$ prompts $p_1, ..., p_n$ with $m$ responses for each $g_1^{p_i}, ..., g_m^{p_i}$, denoted by Met an arbitrary performance metric (e.g., perplexity), $\Delta\text{Met} = \frac{1}{n}\sum_{i=1}^{n}\frac{1}{m}|\sum_{j=1}^{m}\text{Met}(g_j^{p_i}(\text{No watermark})) - \sum_{j=1}^{m}\text{Met}(g_j^{p_i}(\text{Watermarked}))|$

**Weakly Distortion-Free.** The results are presented in Figure 3. This figure shows that compared to the model without watermarks, all weakly distortion-free watermarks exhibit no significant performance bias in text summarization and text generation tasks. However, for the Soft-watermark (Kirchenbauer et al., 2023), a significant performance bias is observable as $\delta$ increases.

**Strongly Distortion-Free.** The results are displayed in Table 2. From this table, it is evident that compared to the baseline, which is the $\Delta$ metrics between two non-watermarked models, all weakly distortion-free watermarks demonstrate performance bias across all tasks. In contrast, the Beta-watermark exhibits less bias compared to other weakly distortion-free watermarks. Additionally, as $\beta$ increases, the distribution bias is further reduced, consistent with our theoretical analysis.

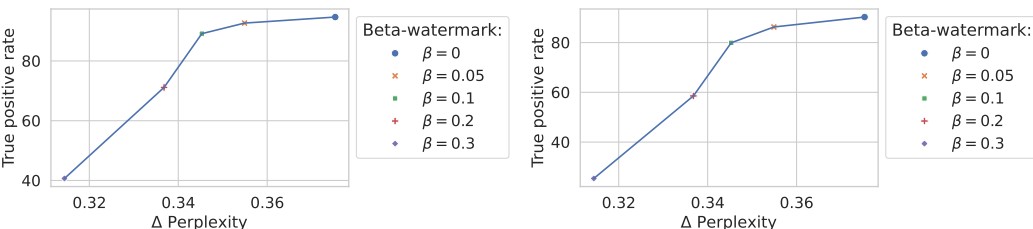

Figure 4: Trade-off between distribution bias and watermark strength under key collision. The TPR is measured under 1% (**Left**), 0.1% (**Right**) FPR. We can see $\Delta$ Perplexity (distribution bias) increase with the TPR.

Table 3: p-value of our black-box distortion-free watermark detection algorithm on text summarization and machine translation tasks. TS: Text Summarization; MT: Machine Translation; IS: Inverse-sampling; GR: Gumble-reparametrization; PR: Permute-reweight. The definition of DetWmk is shown in Sec. 4.3.

| | | IS | GR | PR | Beta watermark ($\beta$) | | | |
|---|---|---|---|---|---|---|---|---|
| | | | | | 0.05 | 0.1 | 0.2 | 0.3 |
| TS | DetWmk | 1.4818 | 1.5864 | 0.0725 | 0.0521 | 0.0425 | 0.0340 | 0.0116 |
| | p-value | 0 | 0 | 0.0125 | 0.1041 | 0.2219 | 0.3816 | 0.8939 |
| MT | DetWmk | 4.7638 | 4.8722 | 0.4597 | 0.4090 | 0.2600 | 0.1795 | 0.0992 |
| | p-value | 0 | 0 | 1.6594e-05 | 1.6450e-04 | 0.0295 | 0.1867 | 0.5989 |

**Trade-off between watermark strength and distribution bias.** We use the beta-watermark to empirically verify the trade-off between watermark strength and distribution bias. As shown in Figure 4, with increasing values of $\beta$, the distribution bias decreases, but there is also a corresponding decrease in the true positive rate of watermark detection. This indicates that reducing the distribution bias of the watermark compromises its detectability.

## 6.2 BLACK-BOX DISTORTION-FREE WATERMARK DETECTION

In this section, we present experimental results to validate our proposed black-box distortion-free watermark detection method (Theorem 4.12). We evaluate the performance on text summarization and machine translation tasks using perplexity and BLEU as the metrics, respectively. To ensure these metrics are bounded, we clip the perplexity to the interval [0,10] and the BLEU score to the interval [0,20]. We report the p-value calculated according to Eq. (1).

From Table 3, we observe that, under a 5% false positive rate (FPR), our detection method successfully identifies inverse-sampling, Gumbel-reparametrization, and permute-reweight watermarks for both text summarization and machine translation tasks. However, the beta-watermark is able to significantly reduce the detection accuracy.

## 7 CONCLUSION

In conclusion, this work identifies three levels of distortion-free capabilities in watermarks—Step-wise, Weakly, and Strongly Distortion-free—and demonstrates that existing watermarks are not strongly distortion-free due to key collisions, which disrupt the original language model distribution across multiple generations. We theoretically establish a trade-off between watermark strength and distribution bias, and introduce a black-box detection approach for identifying watermarked models. Additionally, we prove that strongly distortion-free watermarks are theoretically unattainable. As a practical solution, we propose beta-watermark, a new weakly distortion-free watermark that effectively reduces distribution bias at the cost of watermarking strength. Future research direction includes 1) exploring further details of the trade-off between the distribution bias and the watermarking strengh and 2) developing more efficient watermark detection methods for weakly distortion-free watermarks.

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

## A ALGORITHMS OF BETA-WATERMARK

---

**Algorithm 1** Beta-watermark generator

---

1: **Input:** secret key sk, parameter $\beta$, prompt $\boldsymbol{x}_{-m:0}$, generate length $n \in \mathbb{N}$, texture key history $hist$, n-gram parameter $a$, and permutation generation function $h$.
2: **for** $i = 1, \ldots, n$ **do**
3:     Calculate the LM distribution for generating the $i$-th token $P_M(\cdot \mid \boldsymbol{x}_{-m:i-1})$.
4:     Generate a watermark key $k_i = (\mathsf{sk}, \boldsymbol{x}_{i-a,i-1})$.
5:     **if** $k_i \in hist$ **then**
6:         Sample the next token $x_i$ using original LM distribution $P_M(\cdot \mid \boldsymbol{x}_{-m:i-1})$.
7:     **else**
8:         Generate the permutation of token set $\pi(\cdot | k_i)$.
9:         Calculate watermarked distribution $P_W(\cdot | \boldsymbol{x}_{-m:i-1}) = F_\beta(P_M(\cdot \mid \boldsymbol{x}_{-m:i-1}), k_i)$.
10:        Sample the next token $x_i$ using distribution $P_W(\cdot | \boldsymbol{x}_{-m:i-1})$.
11: **return** $\boldsymbol{x}_{1:n}$.

---

---

**Algorithm 2** Beta-watermark detector

---

1: **Input:** text $\boldsymbol{x}_{1:n}$, secret key sk, volume of the token set $N$, score function $s$, n-gram parameter $a$, threshold $z$.
2: Initialize the score function: $S = 0$.
3: **for** $i = 2, ..., n$ **do**
4:     Generate the watermark key $k_i = (\mathsf{sk}, \boldsymbol{x}_{i-a,i-1})$.
5:     Generate the permutation of token set $\pi(\cdot | k_i)$.
6:     Update the score function via $S = S + s(\pi(x_i | k_i), k_i, F_\beta)$.
7: **return** $S > z\sqrt{n}$.

---

## B RELATED WORK

**Statistical watermarks.** Kirchenbauer et al. (2023) enhanced the statistical watermark framework originally introduced by Aaronson (2022), demonstrating the effectiveness of statistical watermarking through extensive experiments on large language models. They splited the LM tokens into red and green list, then promoted the use of green tokens by adding a fixed parameter $\delta$ to their logits. Zhao et al. (2023) proposed the unigram watermark, which enhances the robustness of the statistical watermark by using one-gram hashing to produce watermark keys. Liu et al. (2023b) also improved the robustness of statistical watermarking by leveraging the semantics of generated content as watermark keys. Additionally, Liu et al. (2023a) proposed an unforgeable watermark scheme that employs neural networks to modify token distributions instead of using traditional watermark keys. However, these approaches may lead to significant changes in the distribution of generated text, potentially compromising content quality.

**Distortion-free watermarks.** To preserve the original output distribution in watermarked content, researchers have explored alternative strategies to modify the token distribution. Aaronson (2022) introduced the first distortion-free watermarking strategy, which utilized Gumbel-reparametrization to alter token distribution and the prefix n-gram content as the watermark keys. Christ et al. (2023) and Kuditipudi et al. (2023) adopted the inverse-sampling and Gumbel-reparametrization to modify the watermarked token distributions, where the watermark keys is based on the token position or a fixed key list respectively. Notice Christ et al. (2023)'s method encounters resilience challenges under modifications and lacks empirical evidence regarding its detectability. Meanwhile, Kuditipudi et al. (2023)'s detection process involves hundreds of resampling steps from the secret key distribution, proving inefficient for processing lengthy texts. Hu et al. (2023) employed inverse-sampling and permute-reweight methods for watermarking. But their detector is not model-agnostic, which requires access to the language model API and prompts, which compromises its operational efficiency.

**Post-hoc Detectors.** Post-hoc detection serves as a significant alternative to watermarking, focusing on the retrospective analysis of machine-generated text. This can be achieved by leveraging inherent features of language models or by enhancing pre-existing expansive models to function as detectors,

as detailed by (Zellers et al., 2019). Specific implementation nuances, such as sampling methods, can be uncovered through reverse engineering the generated text, a process described by (Tay et al., 2020). Additionally, there are post-hoc detectors designed for modern large language models (Mitchell et al., 2023; Tian, 2023; Kirchner et al., 2023), specifically trained for binary detection tasks. However, there is a growing consensus that these detection methods are becoming less effective as language models evolve. As observed by Gambini et al. (2022), detection mechanisms effective against GPT-2 have struggled with GPT-3. Moreover, text rephrasing models like those in (Krishna et al., 2023) are bypassing prevalent post-hoc detectors such as GPTZero (Tian, 2023), DetectGPT (Mitchell et al., 2023), and OpenAI's proprietary detector (Kirchner et al., 2023). Additionally, Chakraborty et al. (2023) notes that as AI-generated content becomes increasingly indistinguishable from human-produced text, the demand on post-hoc detectors to analyze longer text segments will likely increase.

**Steganography.** Steganography involves embedding hidden messages in media such as natural language or images, ensuring only intended recipients can detect the message while it remains concealed from others (Hopper et al., 2002). When applied to watermarking, the goal is to maintain stealth. However, established steganography techniques may not meet this goal without certain entropy-related assumptions. In scenarios where language model prompts can be adversarially chosen, the need for stealth remains. This discrepancy arises due to the different levels of access that watermarking and steganography have to the model's output distribution. In steganography, there is only oracle access to this distribution, whereas our watermarking approach provides a detailed view of the token's probability distribution. Hence, while steganography either depends on entropy assumptions (Hopper et al., 2002) or risks security with low entropy channels (Dedić et al., 2009), our watermark remains stealthy regardless of the text's entropy. This is achieved by leveraging full distribution access and using it as a foundation for embedding watermarks. Kaptchuk et al. (2021) discusses encoding with similar access but presupposes equal decoding access, which is impractical for watermarking as the detection algorithm typically lacks the initiating prompt, thus remaining unaware of the distribution.

## C  DISCUSSION

In this section, we provide detailed discussion of two "undetectable" scheme (Christ et al., 2023; Christ & Gunn, 2024). We claim neither of them can achieve strongly distortion-free.

For the undetectable scheme proposed by Christ & Gunn (2024), it is important to note that strongly distortion-free watermarks require the independence of $F(P_M(x_i \mid x_{1:i-1}), k_i)$ at every generation step $i$. Existing distortion-free watermarks achieve this by using distinct watermark keys $k_i$ and a hash function to ensure independence, which requires the 'key collision' not occuring. In contrast, Christ & Gunn (2024) achieves the independence of $F(P_M(x_i \mid x_{1:i-1}), k_i)$ by developing a key sampling method (termed PRC), which aims for i.i.d. sampling of watermark keys with true randomness, i.e., replacing the pseudorandomness in the PDA-rule with true randomness by randomly sampling the watermark keys. However, despite these efforts, their method still does not achieve strongly distortion-free watermarks, as PRC methods are only close to, but cannot fully achieve, i.i.d. sampling of true randomness (see Lemma 9 in their paper). Therefore, their method can still not achieve strongly distortion-free watermarks.

The undetectable watermark (Christ et al., 2023) is another example of trading detectability for reducing the distribution bias. In Christ et al. (2023), watermarked tokens are produced only if the hash window has an entropy larger than a given threshold $\lambda$, i.e., they skip watermarking the first several tokens to accumulate enough true randomness. However, there is also a trade-off between the watermark strength and the distribution bias under their scheme. This trade-off is controlled by the entropy threshold $\lambda$. When $\lambda$ increases, the number of watermarked tokens decreases, and it will become more difficult to detect the watermark, but the key collision is less likely to occur, and the distribution bias decreases.

For example, if we use Hoeffding's concentration bound as the p-value estimator, i.e., $P(S_n - E[S_n] > s) \leq \exp(-2\frac{s^2}{n})$, when the generated sequence length does not change, the p-value upper bound exponentially increases with the number of non-watermarked tokens (because $s = S_n^{watermarked} - E[S_n]$ is linearly related to the number of watermarked tokens). Thus, although the order of hash windows is $2^\lambda$, the detectability could be the order $O(e^{-\lambda})$ (assuming $\lambda$ is linearly

related to the number of watermarked tokens), which show a trade-off between the distortion bias and the watermark detectability.

Besides, the key sampling methods Alg. 3 and 5 of Christ et al. (2023) are also not resilient to even single text modification. The watermark key space of undetectable watermark is (sk, texture key, position key) (see line 12 of Alg.3, line 7 of Alg.5 and the discussion at the second last paragraph of Section 4.2). The texture key is similar to the definition of n-gram. Since they also use position keys in their watermark key, a single deletion will remove the watermark.

Notice, the texture key in Christ et al. (2023) is similar but not equal to the n-gram hashing. In Alg.3, they use the same texture key during one generation. From my perspective, the texture keys in Christ et al. (2023) are more likely to serve as increasing the diversity of the secret key sk to reduce the distribution-bias in "multiple generation with the same prompt". The diversity of watermark keys in one generation is ensured by the position key.

# D    MISSING PROOFS

## D.1    PROOF OF THEOREM 4.1

*Proof.*

$$
\begin{aligned}
\Pr\left(\max_{i\in[M]}(S(\boldsymbol{x}_{1:n}|\mathsf{sk}_i) - \mathbb{E}_{H_0}[S]) \le t|H_0\right) &= \prod_{i=1}^{M} \Pr\left(S(\boldsymbol{x}_{1:n}|\mathsf{sk}_i) - \mathbb{E}_{H_0}[S] \le t|H_0\right) \\
&= \prod_{i=1}^{M}(1 - \Pr\left(S(\boldsymbol{x}_{1:n}|\mathsf{sk}_i) - \mathbb{E}_{H_0}[S] \ge t|H_0\right)) \\
&= (1 - p_0(t))^M.
\end{aligned}
\tag{2}
$$

Thus,

$$
\begin{aligned}
\Pr\left(\max_{i\in[M]}(S(\boldsymbol{x}_{1:n}|\mathsf{sk}_i) - \mathbb{E}_{H_0}[S]) \ge t|H_0\right) &= 1 - \Pr\left(\max_{i\in[M]}(S(\boldsymbol{x}_{1:n}|\mathsf{sk}_i) - \mathbb{E}_{H_0}[S]) \le t|H_0\right) \\
&= 1 - (1 - p_0(t))^M.
\end{aligned}
\tag{3}
$$

$\square$

## D.2    PROOF OF THEOREM 4.6

*Proof.* We first show the weakly distortion-free case: firstly, if key collision does not occur, we have

$$
\begin{aligned}
\mathbb{E}_{\boldsymbol{k}_{1:n}}[P_W(\boldsymbol{x}_{1:n}|\boldsymbol{k}_{1:n})] &= \mathbb{E}_{\boldsymbol{k}_{1:n}}\left[\prod_{i=1}^{n} F(P_M(x_i|\boldsymbol{x}_{1:i-1}), k_i)\right] \\
&= \prod_{i=1}^{n} \mathbb{E}_{k_i}[F(P_M(x_i|\boldsymbol{x}_{1:i-1}), k_i)].
\end{aligned}
\tag{4}
$$

The above equality stems from the independence property of the PDA-rule $F(P_M(x_i|\boldsymbol{x}_{1:i-1}), k_i)$. Christ et al. (2023) and Hu et al. (2023) show that if there is no repeating keys in $\boldsymbol{k}_{1:n}$, the independence property can be satisfied with hash functions.

Since $F$ is a distortion-free PDA-rule, we have $\mathbb{E}_{k_i}[F(P_M(x_i|\boldsymbol{x}_{1:i-1}), k_i)] = P_M(x_i|\boldsymbol{x}_{1:i-1})$. Thus,

$$
\mathbb{E}_{\boldsymbol{k}_{1:n}}[P_W(\boldsymbol{x}_{1:n}|\boldsymbol{k}_{1:n})] = \prod_{i=1}^{n} \mathbb{E}_{k_i}[F(P_M(x_i|\boldsymbol{x}_{1:i-1}), k_i)] = \prod_{i=1}^{n} P_M(x_i|\boldsymbol{x}_{1:i-1}) = P_M(\boldsymbol{x}_{1:n}). \tag{5}
$$

Analogously, for the strongly distortion-free case, if key collision does not occur, we will have distinct $\boldsymbol{k}_{1:n}^{(i)}$. By the independence property of the PDA-rule, we have

$$
\begin{aligned}
\mathbb{E}_{\boldsymbol{k}_{1:n}^{(1)},\ldots,\boldsymbol{k}_{1:n}^{(N_0)}}[\prod_{i=1}^{N_0} P_W(\boldsymbol{x}_{1:n}^{(i)}|\boldsymbol{k}_{1:n}^{(i)})] &= \prod_{i=1}^{N_0} \mathbb{E}_{\boldsymbol{k}_{1:n}^{(i)}}[P_W(\boldsymbol{x}_{1:n}^{(i)}|\boldsymbol{k}_{1:n}^{(i)})] \\
&= \prod_{i=1}^{N_0}\prod_{j=1}^{n} \mathbb{E}_{k_j^{(i)}}[P_W(x_j^{(i)}|\boldsymbol{x}_{1:j-1}^{(i)}, k_j^{(i)})] \\
&= \prod_{i=1}^{N_0}\prod_{j=1}^{n} \mathbb{E}_{k_j^{(i)}}[F(P_M(x_j^{(i)}|\boldsymbol{x}_{1:j-1}^{(i)}), k_j^{(i)})] \qquad (6) \\
&= \prod_{i=1}^{N_0}\prod_{j=1}^{n} P_M(x_j^{(i)}|\boldsymbol{x}_{0:j-1}^{(i)}) \\
&= \prod_{i=1}^{N_0} P_M(\boldsymbol{x}_{1:n}^{(i)}).
\end{aligned}
$$

$\square$

## D.3   PROOF OF THEOREM 4.9

*Proof.* **Part 1.** We start from proving $\mathbb{D}(P, F_{GR}) = \mathbb{D}(P, F_{IS}) = 1 - \sum_{t \in V} P(t)^2$. Since both $F_{GR}$ and $F_{IS}$ are distortion-free PDA-rule, $P(t) = \mathbb{E}_k[F_{GR}(P|k)(t)] = \mathbb{E}_k[F_{IS}(P|k)(t)]$. Since $F_{GR}(P|k)$ and $F_{IS}(P|k)$ are Dirac distribution, when $F_{GR}(P|k)(t) > 0$, $F_{GR}(P|k)(t) = 1$, and $\mathbb{E}_k[F_{GR}(P|k)(t)] = \mathbb{E}_k[\mathbf{1}_{F_{GR}(P|k)(t)>0}] = \Pr(F_{GR}(P|k)(t) > 0), \forall t \in V$. Thus,

$$
\begin{aligned}
\mathbb{E}_k[\sum_{t \in V} \min\{P(t), F_{GR}(P|k)(t)\}] &= \sum_{t \in V} \mathbb{E}_k[P(t)\mathbf{1}_{F_{GR}(P|k)(t)>0}] \\
&= \sum_{t \in V} \mathbb{E}_k[P(t)\mathbf{1}_{F_{GR}(P|k)(t)>0}] \\
&= \sum_{t \in V} \mathbb{E}_k[P(t)|\mathbf{1}_{F_{GR}(P|k)(t)>0}] \Pr(F_{GR}(P|k)(t) > 0) \qquad (7) \\
&= \sum_{t \in V} P(t)^2.
\end{aligned}
$$

Analogously, $\mathbb{E}_k[\sum_{t \in V} \min\{P(t), F_{IS}(P|k)(t)\}] = \sum_{t \in V} P(t)^2$. Therefore, we have

$$
\mathbb{D}(P, F_{GR}) = \mathbb{D}(P, F_{IS}) = 1 - \sum_{t \in V} P(t)^2. \qquad (8)
$$

**Part 2.** Next, we show $0.5(1 - \max_{t \in V} P(t)) \leq \mathbb{D}(P, F_{PR}) \leq 0.5 - \max\{\max_{t \in V} P(t) - 0.5, 0\}$. Given a permutation on the token list, assume w.l.o.g. the permutation is of order $\{t_1, ..., t_N\}$, in $F_{PR}$ we will arrange the token probabilities on the interval $[0, 1]$ following the permutation order. Denote by $i_0$ the index of the token such that $0.5$ lies in its probability region, then the token probabilities of $\{t_{i_0+1}, ... t_N\}$ will be doubled, while the token probabilities of $\{t_1, ... t_{i_0-1}\}$ will be scaled to 0. Thus, under this permutation,

$$
\sum_{t \in V} \min\{P(t), F_{PR}(P|k)(t)\} = \sum_{i=i_0+1}^{N} P(t_i) + \min\{P(t_{i_0}), 2\xi_{i_0}\},
$$

where $\xi_{i_0}$ is the probability mass of $t_{i_0}$ that is in the interval $[0.5, 1]$, $\max\{P(t_{i_0}) - 0.5, 0\} \leq \xi_{i_0} \leq \min\{0.5, P(t_{i_0})\}$. Next, we consider the reverse permutation $\{t_N, ..., t_1\}$, following the similar discussion, we have

$$
\sum_{t \in V} \min\{P(t), F_{PR}(P|k^r)(t)\} = \sum_{i=1}^{i_0-1} P(t_i) + \min\{P(t_{i_0}), 2(P(t_{i_0}) - \xi_{i_0})\},
$$

where $k^r$ refers the key that lead to the reserved permutation. Thus,

$$\sum_{t \in V} \min\{P(t), F_{PR}(P|k)(t)\} + \sum_{t \in V} \min\{P(t), F_{PR}(P|k^r)(t)\} \tag{9}$$
$$= 1 + \min\{P(t_{i_0}), 2\xi_{i_0}\} + \min\{P(t_{i_0}), 2(P(t_{i_0}) - \xi_{i_0})\} - P(t_{i_0}).$$

Next, we show $P(t_{i_0}) \geq \min\{P(t_{i_0}), 2\xi_{i_0}\} + \min\{P(t_{i_0}), 2(P(t_{i_0}) - \xi_{i_0})\} - P(t_{i_0}) \geq \max\{\max_{t \in V} P(t) - 0.5, 0\}$. The left hand side inequality is trivial, as $\min\{P(t_{i_0}), 2\xi_{i_0}\} + \min\{P(t_{i_0}), 2(P(t_{i_0}) - \xi_{i_0})\} \leq 2P(t_{i_0})$.

For the right hand side inequality, given $\min\{A, 2x\} + \min\{A, 2A - 2x\} = A + \min\{2A - 2x, 2x\}$, we have

$$\min\{P(t_{i_0}), 2\xi_{i_0}\} + \min\{P(t_{i_0}), 2(P(t_{i_0}) - \xi_{i_0})\} - P(t_{i_0}) = 2\min\{P(t_{i_0}) - \xi_{i_0}, \xi_{i_0}\}. \tag{10}$$

Since $0 \leq \max\{P(t_{i_0}) - 0.5, 0\} \leq \xi_{i_0} \leq \min\{0.5, P(t_{i_0})\} \leq P(t_{i_0})$, the minimum value of $\min\{P(t_{i_0}) - \xi_{i_0}, \xi_{i_0}\}$ when $\xi_{i_0}$ take either $\max\{P(t_{i_0}) - 0.5, 0\}$ or $\min\{0.5, P(t_{i_0})\}$, thus

$$\min\{P(t_{i_0}) - \xi_{i_0}, \xi_{i_0}\} \geq \max\{P(t_{i_0}) - 0.5, 0\}. \tag{11}$$

If $P(t_{i_0}) - 0.5 > 0$, it is obvious that $\max_{t \in V} P(t) = P(t_{i_0})$. So

$$\min\{P(t_{i_0}) - \xi_{i_0}, \xi_{i_0}\} \geq \max\{\max_{t \in V} P(t) - 0.5, 0\}. \tag{12}$$

Combining it with Equation 9, we have

$$\sum_{t \in V} \min\{P(t), F_{PR}(P|k)(t)\} + \sum_{t \in V} \min\{P(t), F_{PR}(P|k^r)(t)\} \tag{13}$$
$$= 1 + \min\{P(t_{i_0}), 2\xi_{i_0}\} + \min\{P(t_{i_0}), 2(P(t_{i_0}) - \xi_{i_0})\} - P(t_{i_0}).$$
$$\leq 1 + P(t_{i_0}) \leq 1 + \max_{t \in V} P(t),$$

and

$$\sum_{t \in V} \min\{P(t), F_{PR}(P|k)(t)\} + \sum_{t \in V} \min\{P(t), F_{PR}(P|k^r)(t)\} \tag{14}$$
$$= 1 + \min\{P(t_{i_0}), 2\xi_{i_0}\} + \min\{P(t_{i_0}), 2(P(t_{i_0}) - \xi_{i_0})\} - P(t_{i_0}).$$
$$\geq 1 + 2\max\{\max_{t \in V} P(t) - 0.5, 0\}.$$

Since the permutation over $V$ is uniformly seeded with the watermark keys,

$$\mathbb{D}(P, F_{PR}) = 1 - \mathbb{E}_k[\sum_{t \in V} \min\{P(t), F_{PR}(P|k)(t)\}] \tag{15}$$
$$= 1 - \frac{1}{2}\mathbb{E}_k[\sum_{t \in V} \min\{P(t), F_{PR}(P|k)(t)\} + \sum_{t \in V} \min\{P(t), F_{PR}(P|k^r)(t)\}].$$

Combining it with Equation 13 and Equation 14, we have

$$0.5(1 - \max_{t \in V} P(t)) \leq \mathbb{D}(P, F_{PR}) \leq 0.5 - \max\{\max_{t \in V} P(t) - 0.5, 0\}. \tag{16}$$

**Part 3.** Finally, we show $\mathbb{D}(P, F_{PR}) \leq \mathbb{D}(P, F_{IS}) = \mathbb{D}(P, F_{GR})$. We only need to prove $0.5 - \max\{\max_{t \in V} P(t) - 0.5, 0\} \leq 1 - \sum_{t \in V} P(t)^2$. We have two steps for Part 3.

**Lemma D.1.** *Given $0 \leq x_1, x_2 \leq r_0 \leq r_1$, $x_1 + x_2 = r_1 \leq 1$, we have $x_1^2 + x_2^2 \leq r_0^2 + (r_1 - r_0)^2$.*

**Proof.** $x_1^2 + x_2^2 = x_1^2 + (r_1 - x_1)^2 = 2x_1^2 - 2x_1 r_1 + r_1^2 = 2(x_1 - r_1/2)^2 + r_1^2/2 \leq \min_{x_1} 2(x_1 - r_1/2)^2 + r_1^2/2 = r_0^2 + (r_1 - r_0)^2$.

Thus, by inductive we have $1 - \sum_{t \in V} P(t)^2 \geq 1 - \lfloor \frac{1}{\max_{t \in V} P(t)} \rfloor (\max_{t \in V} P(t))^2 - (1 - \lfloor \frac{1}{\max_{t \in V} P(t)} \rfloor \max_{t \in V} P(t))^2$. Now we continue the proof of the main theorem.

**Step 1.** When $\max_{t \in V} P(t) \geq 0.5$,

$$
1 - \sum_{t \in V} P(t)^2 \geq 1 - (\max_{t \in V} P(t))^2 - (1 - \max_{t \in V} P(t))^2
$$

$$
= 2 \max_{t \in V} P(t) - 2(\max_{t \in V} P(t))^2
$$

$$
= 0.5 - 2(\max_{t \in V} P(t) - 0.5)^2 \qquad (17)
$$

$$
\geq 0.5 - (\max_{t \in V} P(t) - 0.5)
$$

$$
= 0.5 - \max\{\max_{t \in V} P(t) - 0.5, 0\}.
$$

**Step 2.** When $\max_{t \in V} P(t) \leq 0.5$,

$$
1 - \sum_{t \in V} P(t)^2 \geq 1 - \lfloor \frac{1}{\max_{t \in V} P(t)} \rfloor (\max_{t \in V} P(t))^2 - (1 - \lfloor \frac{1}{\max_{t \in V} P(t)} \rfloor \max_{t \in V} P(t))^2
$$

$$
= 2 \lfloor \frac{1}{\max_{t \in V} P(t)} \rfloor \max_{t \in V} P(t) - (\lfloor \frac{1}{\max_{t \in V} P(t)} \rfloor + \lfloor \frac{1}{\max_{t \in V} P(t)} \rfloor^2)(\max_{t \in V} P(t))^2,
$$

$$(18)$$

denote by $\epsilon = \frac{1}{\max_{t \in V} P(t)} - \lfloor \frac{1}{\max_{t \in V} P(t)} \rfloor$, we have $0 \leq \epsilon < 1$ and

$$
1 - \sum_{t \in V} P(t)^2 = 2(\frac{1}{\max_{t \in V} P(t)} - \epsilon) \max_{t \in V} P(t) - ((\frac{1}{\max_{t \in V} P(t)} - \epsilon) + (\frac{1}{\max_{t \in V} P(t)} - \epsilon)^2)(\max_{t \in V} P(t))^2,
$$

$$
= 2 - 2\epsilon \max_{t \in V} P(t) - \left( \max_{t \in V} P(t) - \epsilon \max_{t \in V} P(t)^2 + 1 - 2\epsilon \max_{t \in V} P(t) + \epsilon^2 \max_{t \in V} P(t)^2 \right)
$$

$$
= 1 - \max_{t \in V} P(t) + (\epsilon - \epsilon^2) \max_{t \in V} P(t)^2
$$

$$
\geq 1 - \max_{t \in V} P(t) \geq 0.5 = 0.5 - \max\{\max_{t \in V} P(t) - 0.5, 0\}.
$$

$$(19)$$

By Step 1 and Step 2, we have $\mathbb{D}(P, F_{PR}) \leq \mathbb{D}(P, F_{IS}) = \mathbb{D}(P, F_{GR})$. $\qquad \square$

### D.4 PROOF OF THEOREM 4.10

*Proof.* Consider the scenario of generating multiple responses with the **same-prompt single-token-generation** task. According to Definition 4.5 under the strongly distortion-free condition, one must have $\forall P_M \in \mathcal{P}, \forall N_0 \in \mathbb{N}_+, \forall t^{(i)} \in V, \prod_{i=1}^{N_0} P_M(t^{(i)}) = \mathbb{E}_{k^{(1)}, \dots, k^{(N_0)}}[\prod_{i=1}^{N_0} F(P_M | k^{(i)})(t^{(i)})]$. Under key collisions, there exists at least two $k^{(i)}, k^{(j)}$ are the same. Then we have $\forall P_M \in \mathcal{P}, \exists N_0 \geq 2, \forall t^{(i)} \in V, \prod_{i=1}^{N_0} P_M(t^{(i)}) = \mathbb{E}_k[\prod_{i=1}^{N_0} F(P_M | k)(t^{(i)})]$. We will show that this hold if and only if $\mathbb{D}(P_M, F) = 0$.

**Part 1.** It is obviously that $\mathbb{D}(P_M, F) = 0$ can lead to $\forall N_0 \in \mathbb{N}_+, \forall t^{(i)} \in V, \prod_{i=1}^{N_0} P_M(t^{(i)}) = \mathbb{E}_k[\prod_{i=1}^{N_0} F(P_M | k)(t^{(i)})]$. This is because if $\mathbb{D}(P_M, F) = 0$, $P_M(t^{(i)}) = F(P_M | k)(t^{(i)})$ almost surely and thus $\mathbb{E}_k[\prod_{i=1}^{N_0} F(P_M | k)(t^{(i)})] = \mathbb{E}_k[\prod_{i=1}^{N_0} P_M(t^{(i)})] = \prod_{i=1}^{N_0} P_M(t^{(i)})$.

**Part 2.** Now we will show that if $\exists N_0 \geq 2, \forall t^{(i)} \in V, \prod_{i=1}^{N_0} P_M(t^{(i)}) = \mathbb{E}_k[\prod_{i=1}^{N_0} F(P_M | k)(t^{(i)})]$, then $\mathbb{D}(P_M, F) = 0$.

As $t^{(i)}$ is arbitrary selected, we can choose $t^{(1)} =, \dots, = t^{(N_0)} = t$, then we have $P_M(t)^{N_0} = \mathbb{E}_k[F(P_M | k)(t)^{N_0}]$. By Jensen's inequality, when $N_0 \geq 2$,

$$
P_M(t)^{N_0} = \mathbb{E}_k[F(P_M | k)(t)^{N_0}] \geq (\mathbb{E}_k[F(P_M | k)(t)])^{N_0} = P_M(t)^{N_0}.
$$

The equality is achieved if and only if $F(P_M | k)(t) = \mathbb{E}_k[F(P_M | k)(t)] = P_M(t)$. Thus, $\forall t \in V, \forall k \in K, F(P_M | k)(t) = P_M(t)$, which leads to

$$
\mathbb{D}(P_M, F) = 1 - \mathbb{E}_k[\sum_{t \in V} \min\{P_M(t), F(P_M | k)(t)\}] = 1 - \sum_{t \in V} P_M(t) = 0.
$$

$\qquad \square$

### D.5 PROOF OF THEOREM 4.12

*Proof.* Firstly, let's consider $\frac{1}{m}|\sum_{j=1}^{m}\text{Met}(g_j^{p_i}(P_M)) - \sum_{j=1}^{m}\text{Met}(g_j^{p_i}(P_T))|$, each $\text{Met}(g_j^{p_i}(P_T))$ and $\text{Met}(g_j^{p_i}(P_M))$ are independent distributed. With Hoeffding's inequality we have

$$\Pr(\frac{1}{m}|\sum_{j=1}^{m}\text{Met}(g_j^{p_i}(P_M)) - \sum_{j=1}^{m}\text{Met}(g_j^{p_i}(P_T))| > t) < e^{-\frac{mt^2}{2A^2}}.$$

Denote by $\text{DetWmk}_i(P_M, P_T) = \frac{1}{m}|\sum_{j=1}^{m}\text{Met}(g_j^{p_i}(P_M)) - \sum_{j=1}^{m}\text{Met}(g_j^{p_i}(P_T))| - \frac{1}{m}|\sum_{j=1}^{m}\text{Met}(g_j^{p_i}(P_M)) - \sum_{j=1}^{m}\text{Met}(g_j^{p_i}(P_{M'}))|$. Since if $\text{DetWmk}_i(P_M, P_T) > t$, we must have $\frac{1}{m}|\sum_{j=1}^{m}\text{Met}(g_j^{p_i}(P_M)) - \sum_{j=1}^{m}\text{Met}(g_j^{p_i}(P_T))| > t$, which yields

$$\Pr(\text{DetWmk}_i(P_M, P_T) > t) \le \Pr(\frac{1}{m}|\sum_{j=1}^{m}\text{Met}(g_j^{p_i}(P_M)) - \sum_{j=1}^{m}\text{Met}(g_j^{p_i}(P_T))| > t) < e^{-\frac{mt^2}{2A^2}}.$$

$$(20)$$

Analogously, we have $\Pr(\text{DetWmk}_i(P_M, P_T) < -t) < e^{-\frac{mt^2}{2A^2}}$ Thus, $\text{DetWmk}_i(P_M, P_T)$ is sub-Gaussian distributed and it is easy to observe that $\mathbb{E}[\text{DetWmk}_i(P_M, P_T)] = 0$. Since $\text{DetWmk}_i(P_M, P_T), i = 1, ..., n$ is independently distributed, applying Hoeffding's inequality again, we have

$$\Pr(\text{DetWmk}(P_M, P_T) > t) < \exp(-\frac{mn^2t^2}{2A^2\sum_{i=1}^{n}||\text{DetWmk}_i(P_M, P_T)||_{\psi_2}^2}),$$

where $||X||_{\psi_2} = \inf\{c > 0 : \mathbb{E}[e^{X^2/c^2}] \le 2\}$.

Now we need to calculate $||\text{DetWmk}_i(P_M, P_T)||_{\psi_2}^2$. We start from calculating $\mathbb{E}[e^{\text{DetWmk}_i(P_M,P_T)^2/c^2}]$. Since $|\text{DetWmk}_i(P_M, P_T)| \le A$ and the probability density function of $\text{DetWmk}_i(P_M, P_T)$ is symmetric with respect to 0, we have

$$\mathbb{E}[e^{\text{DetWmk}_i(P_M,P_T)^2/c^2}] = \int_{-A}^{A} e^{x^2/c^2} dF_{\text{DetWmk}_i(P_M,P_T)}(x)$$

$$= e^{x^2/c^2} F_{\text{DetWmk}_i(P_M,P_T)}(x)|_{i=-A}^{A} - \int_{-A}^{A} \frac{2x}{c^2} e^{x^2/c^2} F_{\text{DetWmk}_i(P_M,P_T)}(x) dx$$

$$= e^{A^2/c^2} - \int_{-A}^{A} \frac{2x}{c^2} e^{x^2/c^2} (1 - \Pr(\text{DetWmk}_i(P_M, P_T) > x)) dx$$

$$= e^{A^2/c^2} + \int_{-A}^{A} \frac{2x}{c^2} e^{x^2/c^2} \Pr(\text{DetWmk}_i(P_M, P_T) > x) dx$$

$$= e^{A^2/c^2} + \int_{-A}^{0} \frac{2x}{c^2} e^{x^2/c^2} \Pr(\text{DetWmk}_i(P_M, P_T) > x) dx$$

$$+ \int_{0}^{A} \frac{2x}{c^2} e^{x^2/c^2} \Pr(\text{DetWmk}_i(P_M, P_T) > x) dx$$

$$= e^{A^2/c^2} - \int_{0}^{A} \frac{2x}{c^2} e^{x^2/c^2} (1 - \Pr(\text{DetWmk}_i(P_M, P_T) < -x)) dx$$

$$+ \int_{0}^{A} \frac{2x}{c^2} e^{x^2/c^2} \Pr(\text{DetWmk}_i(P_M, P_T) > x) dx$$

$$= e^{A^2/c^2} - \int_{0}^{A} \frac{2x}{c^2} e^{x^2/c^2} dx + 2\int_{0}^{A} \frac{2x}{c^2} e^{x^2/c^2} \Pr(\text{DetWmk}_i(P_M, P_T) > x) dx$$

$$= 1 + 2\int_{0}^{A} \frac{2x}{c^2} e^{x^2/c^2} \Pr(\text{DetWmk}_i(P_M, P_T) > x) dx$$

$$(21)$$

Since when $x \geq 0$, $\Pr(\text{DetWmk}_i(P_M, P_T) > x) \leq e^{-\frac{mt^2}{2A^2}}$, we have

$$
\begin{aligned}
\mathbb{E}[e^{\text{DetWmk}_i(P_M,P_T)^2/c^2}] &= 1 + 2 \int_0^A \frac{2x}{c^2} e^{x^2/c^2} \Pr(\text{DetWmk}_i(P_M, P_T) > x) dx \\
&\leq 1 + 2 \int_0^A \frac{2x}{c^2} e^{x^2/c^2} e^{-\frac{mt^2}{2A^2}} dx \\
&= 1 + \frac{2}{\frac{mc^2}{2A^2} - 1}(1 - e^{A^2/c^2 - m/2})
\end{aligned}
\tag{22}
$$

Taking $c = \sqrt{\frac{6A^2}{m}}$, $\mathbb{E}[e^{\text{DetWmk}_i(P_M,P_T)^2/c^2}] \leq 1 + (1 - e^{-m/3}) < 2$. Thus, $\|\text{DetWmk}_i(P_M, P_T)\|^2_{\psi_2} \leq \frac{6A^2}{m}$ and

$$
\Pr(\text{DetWmk}(P_M, P_T) > t) < \exp(-\frac{mn^2t^2}{2A^2 \sum_{i=1}^n \|\text{DetWmk}_i(P_M, P_T)\|^2_{\psi_2}}) < \exp(-\frac{m^2nt^2}{12A^4}),
$$

$\square$

### D.6 Proof of Theorem 4.13

*Proof.* Combining Lemma D.2 and Lemma D.3 yields the result.

**Lemma D.2.** *A strongly distortion-free watermark must use independent hashed watermark keys $h(k)$.*

*Proof.* The proof of this lemma is similar to the proof of Theorem 4.10. We prove by contradiction, if we don't have independent hashed watermark keys $h(k)$, then given two randomly sampled key $h(k_1)$ and $h(k_2)$, $\Pr(h(k_2) = A, h(k_1) = B) \neq \Pr(h(k_1) = A) \Pr(h(k_2) = B)$. Consider the scenario of generating multiple responses with the **same-prompt one-token-generation** task. According to Definition 4.5 under the strongly distortion-free condition, one must have $\forall P_M \in \mathcal{P}, \forall N_0 \in \mathbb{N}_+, \forall t^{(i)} \in V, \prod_{i=1}^{N_0} P_M(t^{(i)}) = \mathbb{E}_{h(k^{(1)}),...,h(k^{(N_0)})}[\prod_{i=1}^{N_0} F(P_M|h(k^{(i)}))(t^{(i)})]$.

We show that if $\exists N_0 \geq 2, \forall t^{(i)} \in V, \prod_{i=1}^{N_0} P_M(t^{(i)}) = \mathbb{E}_{h(k^{(1)}),...,h(k^{(N_0)})}[\prod_{i=1}^{N_0} F(P_M|h(k^{(i)}))(t^{(i)})]$, then $\mathbb{D}(P_M, F) = 0$.

As $t^{(i)}$ is arbitrary selected, we can choose $t^{(1)} =, ..., = t^{(N_0)} = t$, then we have $P_M(t)^{N_0} = \mathbb{E}_{h(k^{(1)}),...,h(k^{(N_0)})}[\prod_{i=1}^{N_0} F(P_M|h(k^{(i)}))(t)]$. Assume w.l.o.g. $N_0 \geq 2$,

$$
\begin{aligned}
&P_M(t)^2 - \mathbb{E}_{h(k^{(1)}),h(k^{(2)})}[F(P_M|h(k^{(1)}))(t)F(P_M|h(k^{(2)}))(t)], \\
=&P_M(t)^2 - \sum_A \sum_B F(P_M|A)(t)F(P_M|B)(t) \Pr(h(k^{(1)}) = A, h(k^{(2)}) = B), \\
=&\sum_A \sum_B F(P_M|A)(t)F(P_M|B)(t) \Pr(h(k^{(1)}) = A) \Pr(h(k^{(2)}) = B) \\
&-\sum_A \sum_B F(P_M|A)(t)F(P_M|B)(t) \Pr(h(k^{(1)}) = A, h(k^{(2)}) = B), \\
=&\sum_A \sum_B F(P_M|A)(t)F(P_M|B)(t)[\Pr(h(k^{(1)}) = A) \Pr(h(k^{(2)}) = B) - \Pr(h(k^{(1)}) = A, h(k^{(2)}) = B)].
\end{aligned}
\tag{23}
$$

Since $\exists A, B$, such that $\Pr(h(k^{(1)}) = A) \Pr(h(k^{(2)}) = B) \neq \Pr(h(k^{(1)}) = A, h(k^{(2)}) = B)$, there exists a $P_M$ such that

$$
\begin{aligned}
&\sum_A \sum_B F(P_M|A)(t)F(P_M|B)(t)[\Pr(h(k^{(1)}) = A) \Pr(h(k^{(2)}) = B) - \Pr(h(k^{(1)}) = A, h(k^{(2)}) = B)]_+ \\
&+\sum_A \sum_B F(P_M|A)(t)F(P_M|B)(t)[\Pr(h(k^{(1)}) = A) \Pr(h(k^{(2)}) = B) - \Pr(h(k^{(1)}) = A, h(k^{(2)}) = B)]_- \neq 0.
\end{aligned}
\tag{24}
$$

In this case, $P_M(t)^2 - \mathbb{E}_{h(k^{(1)}), h(k^{(2)})}[F(P_M|h(k^{(1)}))(t)F(P_M|h(k^{(2)}))(t)] \neq 0$, thus the watermark is not strongly distortion-free. $\qquad\square$

**Lemma D.3.** *A watermark using a distortion-free and independent PDA-rule is undetectable by any arbitrary detector.*

*Proof.* Recall that the watermarking detection algorithm utilize the statistical difference between the watermarked LM and the original LM to check the existence of the watermark, i.e., the detection is based on $\mathbb{E}[P_M(\boldsymbol{x}_{1:n})|\text{DetCon}] \neq \mathbb{E}[\prod_{i=1}^{n} F(P_M|h(k^{(i)}))(t^{(i)})|\text{DetCon}]$, where DetCon is the detecting condition which is used in watermark generator. Now we show if the PDA-rule is independent from each other, the DetCon will be independent of the PDA-rule. This can be shown by contradiction.

If DetCon is not independent of the PDA-rules, during the generation process, the PDA-rules will be mutually dependent because they all share dependency on the DetCon, which contradicts to the independence of PDA-rule. Thus, we have

$$\mathbb{E}[\prod_{i=1}^{n} F(P_M|h(k^{(i)}))(t^{(i)})|\text{DetCon}] = \prod_{i=1}^{n} \mathbb{E}[F(P_M|h(k^{(i)}))(t^{(i)})|\text{DetCon}] = \mathbb{E}[P_M(\boldsymbol{x}_{1:n})|\text{DetCon}].$$

Thus, the watermark is undetectable by the detector. $\qquad\square$

$\qquad\square$

### D.7 PROOF OF THEOREM 5.2

*Proof.* We need to show $P_M(t|\boldsymbol{x}_{1:i-1}) = \mathbb{E}_{k_i}[F_\beta(P_M(\cdot|\boldsymbol{x}_{1:i-1}), k_i)(t)]$. As $F_{PR}(P_M(\cdot|\boldsymbol{x}_{1:i-1}), k_i)(t)$ is a distortion-free PDA-rule, we know $\mathbb{E}_{k_i}[(1 - \beta)F_{PR}(P_M(\cdot|\boldsymbol{x}_{1:i-1}), k_i)(t)] = (1 - \beta)P_M(t|\boldsymbol{x}_{1:i-1})$. Thus, we need to show

$$\mathbb{E}_{k_i}\left[\max\{2\sum_{t',\pi(t'|k_i)\geq\pi(t|k_i)} P_M(t'|\boldsymbol{x}_{1:i-1}) - 1, 0\} - \max\{2\sum_{t',\pi(t'|k_i)\geq\pi(t|k_i)+1} P_M(t'|\boldsymbol{x}_{1:i-1}) - 1, 0\}\right]$$

$$= P_M(t|\boldsymbol{x}_{1:i-1}) \tag{25}$$

Since the permutation is uniformly distributed, denoted by $\Pi$ the set of all permutations on $V$ and $P_\Pi$ the uniformly distribution on $\Pi$, we have

$$\mathbb{E}_{k_i}\left[\max\{2\sum_{t',\pi(t'|k_i)\geq\pi(t|k_i)} P_M(t'|\boldsymbol{x}_{1:i-1}) - 1, 0\} - \max\{2\sum_{t',\pi(t'|k_i)\geq\pi(t|k_i)+1} P_M(t'|\boldsymbol{x}_{1:i-1}) - 1, 0\}\right]$$

$$= \mathbb{E}_{\pi\sim P_\Pi}\left[\max\{2\sum_{t',\pi(t'|k_i)\geq\pi(t|k_i)} P_M(t'|\boldsymbol{x}_{1:i-1}) - 1, 0\} - \max\{2\sum_{t',\pi(t'|k_i)\geq\pi(t|k_i)+1} P_M(t'|\boldsymbol{x}_{1:i-1}) - 1, 0\}\right] \tag{26}$$

As $P_\Pi$ is the uniformly distribution on $\Pi$, for each $\pi \in \Pi$, we consider its reverse permutation $\pi^r$:

$$\mathbb{E}_{\pi\sim P_\Pi}\left[\max\{2\sum_{t',\pi(t'|k_i)\geq\pi(t|k_i)} P_M(t'|\boldsymbol{x}_{1:i-1}) - 1, 0\} - \max\{2\sum_{t',\pi(t'|k_i)\geq\pi(t|k_i)+1} P_M(t'|\boldsymbol{x}_{1:i-1}) - 1, 0\}\right]$$

$$= \frac{1}{2}\mathbb{E}_{\pi^r\sim P_\Pi}\left[\max\{2\sum_{t',\pi(t'|k_i)\geq\pi(t|k_i)} P_M(t'|\boldsymbol{x}_{1:i-1}) - 1, 0\} - \max\{2\sum_{t',\pi(t'|k_i)\geq\pi(t|k_i)+1} P_M(t'|\boldsymbol{x}_{1:i-1}) - 1, 0\}\right.$$

$$\left. + \max\{2\sum_{t',\pi^r(t'|k_i)\geq\pi^r(t|k_i)} P_M(t'|\boldsymbol{x}_{1:i-1}) - 1, 0\} - \max\{2\sum_{t',\pi^r(t'|k_i)\geq\pi^r(t|k_i)+1} P_M(t'|\boldsymbol{x}_{1:i-1}) - 1, 0\}\right] \tag{27}$$

Notice, if $\pi(t') \leq \pi(t)$, then in the reversed permutation $\pi^r$, we have $\pi^r(t') \geq \pi^r(t)$ and vice versa. Thus,

$$
\max\{2 \sum_{t',\pi^r(t'|k_i)\geq\pi^r(t|k_i)} P_M(t'|\boldsymbol{x}_{1:i-1}) - 1, 0\} - \max\{2 \sum_{t',\pi^r(t'|k_i)\geq\pi^r(t|k_i)+1} P_M(t'|\boldsymbol{x}_{1:i-1}) - 1, 0\}
$$

$$
= \max\{2 \sum_{t',\pi(t'|k_i)\leq\pi(t|k_i)} P_M(t'|\boldsymbol{x}_{1:i-1}) - 1, 0\} - \max\{2 \sum_{t',\pi(t'|k_i)\leq\pi(t|k_i)-1} P_M(t'|\boldsymbol{x}_{1:i-1}) - 1, 0\}
$$

$$
= \max\{1 - 2 \sum_{t',\pi(t'|k_i)\geq\pi(t|k_i)+1} P_M(t'|\boldsymbol{x}_{1:i-1}), 0\} - \max\{1 - 2 \sum_{t',\pi(t'|k_i)\geq\pi(t|k_i)-1} P_M(t'|\boldsymbol{x}_{1:i-1}), 0\}.
$$

$$(28)$$

By $\max\{x,0\} - \max\{-x,0\} = x$ we have

$$
\mathbb{E}_{\pi\sim P_\Pi}\left[\max\{2 \sum_{t',\pi(t'|k_i)\geq\pi(t|k_i)} P_M(t'|\boldsymbol{x}_{1:i-1}) - 1, 0\} - \max\{2 \sum_{t',\pi(t'|k_i)\geq\pi(t|k_i)+1} P_M(t'|\boldsymbol{x}_{1:i-1}) - 1, 0\}\right]
$$

$$
= \frac{1}{2}\mathbb{E}_{\pi\sim P_\Pi}\left[\max\{2 \sum_{t',\pi(t'|k_i)\geq\pi(t|k_i)} P_M(t'|\boldsymbol{x}_{1:i-1}) - 1, 0\} - \max\{2 \sum_{t',\pi(t'|k_i)\geq\pi(t|k_i)+1} P_M(t'|\boldsymbol{x}_{1:i-1}) - 1, 0\}\right.
$$

$$
\left. + \max\{2 \sum_{t',\pi^r(t'|k_i)\geq\pi^r(t|k_i)} P_M(t'|\boldsymbol{x}_{1:i-1}) - 1, 0\} - \max\{2 \sum_{t',\pi^r(t'|k_i)\geq\pi^r(t|k_i)+1} P_M(t'|\boldsymbol{x}_{1:i-1}) - 1, 0\}\right]
$$

$$
= \frac{1}{2}\mathbb{E}_{\pi\sim P_\Pi}\left[\max\{2 \sum_{t',\pi(t'|k_i)\geq\pi(t|k_i)} P_M(t'|\boldsymbol{x}_{1:i-1}) - 1, 0\} - \max\{2 \sum_{t',\pi(t'|k_i)\geq\pi(t|k_i)+1} P_M(t'|\boldsymbol{x}_{1:i-1}) - 1, 0\}\right.
$$

$$
\left. + \max\{1 - 2 \sum_{t',\pi(t'|k_i)\geq\pi(t|k_i)+1} P_M(t'|\boldsymbol{x}_{1:i-1}), 0\} - \max\{1 - 2 \sum_{t',\pi(t'|k_i)\geq\pi(t|k_i)-1} P_M(t'|\boldsymbol{x}_{1:i-1}), 0\}\right]
$$

$$
= \frac{1}{2}\mathbb{E}_{\pi\sim P_\Pi}[2 \sum_{t',\pi^r(t'|k_i)\geq\pi^r(t|k_i)} P_M(t'|\boldsymbol{x}_{1:i-1}) - 1 - (2 \sum_{t',\pi^r(t'|k_i)\geq\pi^r(t|k_i)+1} P_M(t'|\boldsymbol{x}_{1:i-1}) - 1)]
$$

$$
= \frac{1}{2}\mathbb{E}_{\pi\sim P_\Pi}[2P_M(t|\boldsymbol{x}_{1:i-1})]
$$

$$
= P_M(t|\boldsymbol{x}_{1:i-1}).
$$

$$(29)$$

$\square$

### D.8 PROOF OF THEOREM 5.4

*Proof.* **Part 1.** We first show $\forall P \in \mathcal{P}, \mathbb{D}(P, F_\beta) \leq \mathbb{D}(P, F_{PR}) - \beta(1 - \max_{t\in V} P(t))$. According to the Part 2 of Proof D.3, we know that given a permutation $\{t_1, ..., t_N\}$ and let $t_{i_0}$ is the token whose probability mass expands across $1/2$,

$$
\sum_{t\in V} \min\{P(t), F_{PR}(P|k)(t)\} = \sum_{i=i_0+1}^{N} P(t_i) + \min\{P(t_{i_0}), 2\xi_{i_0}\},
$$

where $\xi_{i_0}$ is the probability mass of $t_{i_0}$ that is in the interval $[0.5, 1]$ (notice $t_{i_0}$ is the same for both permuta-reweight and beta PDA-rule as they use the same permutation), $\max\{P(t_{i_0}) - 0.5, 0\} \leq \xi_{i_0} \leq \min\{0.5, P(t_{i_0})\}$. And

$$
\sum_{t\in V} \min\{P(t), F_{PR}(P|k^r)(t)\} = \sum_{i=1}^{i_0-1} P(t_i) + \min\{P(t_{i_0}), 2(P(t_{i_0}) - \xi_{i_0})\},
$$

where $k^r$ refers the key that lead to the reserved permutation.

Now we consider $F_\beta$. From the similar analysis, we have

$$\sum_{t \in V} \min\{P(t), F_\beta(P|k)(t)\} = \sum_{i=i_0+1}^{N} P(t_i) + 2\beta \sum_{i=1}^{i_0-1} P(t_i) + \min\{P(t_{i_0}), 2(1-\beta)\xi_{i_0} + 2\beta(P(t_{i_0})-\xi_{i_0})\},$$

and

$$\sum_{t \in V} \min\{P(t), F_\beta(P|k^r)(t)\} = \sum_{i=1}^{i_0-1} P(t_i) + 2\beta \sum_{i=i_0+1}^{N} P(t_i) + \min\{P(t_{i_0}), 2(1-\beta)(P(t_{i_0})-\xi_{i_0})+2\beta\xi_{i_0}\}.$$

As

$$\min\{P(t_{i_0}), 2(1-\beta)\xi_{i_0} + 2\beta(P(t_{i_0})-\xi_{i_0})\} + \min\{P(t_{i_0}), 2(1-\beta)(P(t_{i_0})-\xi_{i_0}) + 2\beta\xi_{i_0}\}$$
$$= P(t_{i_0}) + \min\{2(1-\beta)(P(t_{i_0})-\xi_{i_0}) + 2\beta\xi_{i_0}, 2(1-\beta)\xi_{i_0} + 2\beta(P(t_{i_0})-\xi_{i_0})\}$$
$$= P(t_{i_0}) + 2\xi_{i_0} + \min\{2(1-\beta)(P(t_{i_0})-2\xi_{i_0}), 2\beta(P(t_{i_0})-2\xi_{i_0})\}$$
$$\geq P(t_{i_0}) + 2\xi_{i_0} + \min\{0, 2(P(t_{i_0})-2\xi_{i_0})\}$$
$$= \min\{P(t_{i_0}), 2(P(t_{i_0})-\xi_{i_0})\} + \min\{P(t_{i_0}), 2\xi_{i_0}\}, \tag{30}$$

we have

$$\sum_{t \in V} \min\{P(t), F_\beta(P|k)(t)\} + \sum_{t \in V} \min\{P(t), F_\beta(P|k^r)(t)\}$$
$$= 1 - P(t_0) + 2\beta(1 - P(t_0)) + \min\{P(t_{i_0}), 2(1-\beta)\xi_{i_0} + 2\beta(P(t_{i_0})-\xi_{i_0})\}$$
$$+ \min\{P(t_{i_0}), 2(1-\beta)(P(t_{i_0})-\xi_{i_0}) + 2\beta\xi_{i_0}\} \tag{31}$$
$$\geq \sum_{t \in V} \min\{P(t), F_{PR}(P|k)(t)\} + \sum_{t \in V} \min\{P(t), F_{PR}(P|k^r)(t)\} + 2\beta - 2\beta P(t_{i_0})$$
$$\geq \sum_{t \in V} \min\{P(t), F_{PR}(P|k)(t)\} + \sum_{t \in V} \min\{P(t), F_{PR}(P|k^r)(t)\} + 2\beta - 2\beta \max_{t \in V} P(t).$$

Thus,

$$\mathbb{D}(P, F_\beta) = 1 - \mathbb{E}_k[\sum_{t \in V} \min\{P(t), F_\beta(P|k)(t)\}]$$

$$= 1 - \frac{1}{2}\mathbb{E}_k[\sum_{t \in V} \min\{P(t), F_\beta(P|k)(t)\} + \sum_{t \in V} \min\{P(t), F_\beta(P|k^r)(t)\}]$$

$$\leq 1 - \frac{1}{2}\mathbb{E}_k[\sum_{t \in V} \min\{P(t), F_{PR}(P|k)(t)\} + \sum_{t \in V} \min\{P(t), F_{PR}(P|k^r)(t)\} + 2\beta - 2\beta \max_{t \in V} P(t)]$$

$$= \mathbb{D}(P, F_{PR}) - \beta(1 - \max_{t \in V} P(t)). \tag{32}$$

**Part 2.** We then show $\forall P \in \mathcal{P}$, if $\beta_1 \leq \beta_2$, then $\mathbb{D}(P, F_{\beta_1}) \geq \mathbb{D}(P, F_{\beta_2})$. Consider $\mathbb{D}(P, F_{\beta_1}) - \mathbb{D}(P, F_{\beta_2})$, we have

$$\mathbb{D}(P, F_{\beta_1}) - \mathbb{D}(P, F_{\beta_2})$$
$$= \mathbb{E}_k[\sum_{t \in V} \min\{P(t), F_{\beta_2}(P|k)(t)\}] - \mathbb{E}_k[\sum_{t \in V} \min\{P(t), F_{\beta_1}(P|k)(t)\}]$$
$$= \frac{1}{2}\mathbb{E}_k\left[\sum_{t \in V} \min\{P(t), F_{\beta_2}(P|k)(t)\} + \sum_{t \in V} \min\{P(t), F_{\beta_2}(P|k^r)(t)\} \right. \tag{33}$$
$$\left. - \sum_{t \in V} \min\{P(t), F_{\beta_1}(P|k)(t)\} - \sum_{t \in V} \min\{P(t), F_{\beta_1}(P|k^r)(t)\}\right]$$

From the similar analysis as Part 1 we have for $F_{\beta_1}$,

$$\sum_{t \in V} \min\{P(t), F_{\beta_1}(P|k)(t)\} = \sum_{i=i_0+1}^{N} P(t_i) + 2\beta_1 \sum_{i=1}^{i_0-1} P(t_i) + \min\{P(t_{i_0}), 2(1-\beta_1)\xi_{i_0} + 2\beta_1(P(t_{i_0})-\xi_{i_0})\},$$

and

$$\sum_{t \in V} \min\{P(t), F_{\beta_1}(P|k^r)(t)\} = \sum_{i=1}^{i_0-1} P(t_i) + 2\beta_1 \sum_{i=i_0+1}^{N} P(t_i) + \min\{P(t_{i_0}), 2(1-\beta_1)(P(t_{i_0}) - \xi_{i_0}) + 2\beta_1\xi_{i_0}\}.$$

For $F_{\beta_2}$, we have

$$\sum_{t \in V} \min\{P(t), F_{\beta_2}(P|k)(t)\} = \sum_{i=i_0+1}^{N} P(t_i) + 2\beta_2 \sum_{i=1}^{i_0-1} P(t_i) + \min\{P(t_{i_0}), 2(1-\beta_2)\xi_{i_0} + 2\beta_2(P(t_{i_0}) - \xi_{i_0})\},$$

and

$$\sum_{t \in V} \min\{P(t), F_{\beta_2}(P|k^r)(t)\} = \sum_{i=1}^{i_0-1} P(t_i) + 2\beta_2 \sum_{i=i_0+1}^{N} P(t_i) + \min\{P(t_{i_0}), 2(1-\beta_2)(P(t_{i_0}) - \xi_{i_0}) + 2\beta_2\xi_{i_0}\}.$$

When $\beta_2 \geq \beta_1$,

$$\min\{P(t_{i_0}), 2(1-\beta_2)\xi_{i_0} + 2\beta_2(P(t_{i_0}) - \xi_{i_0})\} + \min\{P(t_{i_0}), 2(1-\beta_2)(P(t_{i_0}) - \xi_{i_0}) + 2\beta_2\xi_{i_0}\}$$
$$= P(t_{i_0}) + \min\{2(1-\beta_2)(P(t_{i_0}) - \xi_{i_0}) + 2\beta_2\xi_{i_0}, 2(1-\beta_2)\xi_{i_0} + 2\beta_2(P(t_{i_0}) - \xi_{i_0})\}$$
$$= P(t_{i_0}) + 2\xi_{i_0} + \min\{2(1-\beta_2)(P(t_{i_0}) - 2\xi_{i_0}), 2\beta_2(P(t_{i_0}) - 2\xi_{i_0})\}$$
$$\geq P(t_{i_0}) + 2\xi_{i_0} + \min\{2(1-\beta_1)(P(t_{i_0}) - 2\xi_{i_0}), 2\beta_1(P(t_{i_0}) - 2\xi_{i_0})\}$$
$$= \min\{P(t_{i_0}), 2(1-\beta_1)\xi_{i_0} + 2\beta_1(P(t_{i_0}) - \xi_{i_0})\} + \min\{P(t_{i_0}), 2(1-\beta_1)(P(t_{i_0}) - \xi_{i_0}) + 2\beta_1\xi_{i_0}\}.$$
$$(34)$$

Thus,

$$\sum_{t \in V} \min\{P(t), F_{\beta_2}(P|k)(t)\} + \sum_{t \in V} \min\{P(t), F_{\beta_2}(P|k^r)(t)\}$$
$$= 1 - P(t_0) + 2\beta_2(1 - P(t_0)) + \min\{P(t_{i_0}), 2(1-\beta_2)\xi_{i_0} + 2\beta_2(P(t_{i_0}) - \xi_{i_0})\}$$
$$+ \min\{P(t_{i_0}), 2(1-\beta_2)(P(t_{i_0}) - \xi_{i_0}) + 2\beta_2\xi_{i_0}\}$$
$$\geq \sum_{t \in V} \min\{P(t), F_{\beta_1}(P|k)(t)\} + \sum_{t \in V} \min\{P(t), F_{\beta_1}(P|k^r)(t)\} + 2(\beta_2 - \beta_1)(1 - P(t_{i_0}))$$
$$\geq \sum_{t \in V} \min\{P(t), F_{\beta_1}(P|k)(t)\} + \sum_{t \in V} \min\{P(t), F_{\beta_1}(P|k^r)(t)\} + 2(\beta_2 - \beta_1)(1 - \max_{t \in V} P(t)).$$
$$(35)$$

Combining with Equation 33 we have:
$$\mathbb{D}(P, F_{\beta_1}) - \mathbb{D}(P, F_{\beta_2})$$
$$= \mathbb{E}_k[\sum_{t \in V} \min\{P(t), F_{\beta_2}(P|k)(t)\}] - \mathbb{E}_k[\sum_{t \in V} \min\{P(t), F_{\beta_1}(P|k)(t)\}]$$
$$= \frac{1}{2}\mathbb{E}_k\left[\sum_{t \in V} \min\{P(t), F_{\beta_2}(P|k)(t)\} + \sum_{t \in V} \min\{P(t), F_{\beta_2}(P|k^r)(t)\}\right.$$
$$\left. - \sum_{t \in V} \min\{P(t), F_{\beta_1}(P|k)(t)\} - \sum_{t \in V} \min\{P(t), F_{\beta_1}(P|k^r)(t)\}\right] \quad (36)$$
$$\geq (\beta_2 - \beta_1)(1 - \max_{t \in V} P(t)) \geq 0.$$

Therefore, $\mathbb{D}(P, F_{\beta_1}) \geq \mathbb{D}(P, F_{\beta_2})$. $\qquad\square$

### D.9 PROOF OF DEFINITION 5.5

*Proof.* We prove the concentration bound in Definition 5.5: $\Pr(S(\boldsymbol{x}_{1:n}) - \mathbb{E}_{H_0}[S(\boldsymbol{x}_{1:n})] > t\sqrt{n}|H_0) \leq \exp(-2t^2)$. Since the range of the sigmoid function is in $[0,1]$, by Hoeffding's inequality, for each random score $s(x_i)$, we have

$$\Pr(\frac{1}{n}\sum_{i=1}^{n} s(x_i) - \mathbb{E}_{H_0}[\frac{1}{n}\sum_{i=1}^{n} s(x_i)] > t|H_0) \leq e^{-2nt^2}. \quad (37)$$

Replace $t$ by $\frac{t}{\sqrt{n}}$ we have

$$\Pr(\sum_{i=1}^{n} s(x_i) - \mathbb{E}_{H_0}[\sum_{i=1}^{n} s(x_i)] > t\sqrt{n}|H_0) \leq e^{-2t^2}. \tag{38}$$

$\square$

## E  DETAILED EXPERIMENT SETUP

### E.1  EXPERIMENT SETUP

We evaluate the distortion-free performance of various watermark models within two seq2seq applications: text summarization and text generation. The experiments leverage the Huggingface library (Wolf et al., 2019), a popular framework for model development and sharing in the NLP community. All tests are conducted on 8 NVIDIA A6000 GPUs, each with 48GB of memory.

We focus on three seq2seq tasks in our experiments: machine translation, text summarization and text generation. For the machine translation task, we focus on English-to-Romanian translation. We employ the Multilingual BART (MBart) model (Liu et al., 2020) on the WMT'14 En-Ro corpus. For text summarization, we employ the BART-large model (Liu et al., 2020) using the CNN-DM corpus dataset (Hermann et al., 2015). For text generation, we follow the settings described by (Kirchenbauer et al., 2023), using the LLaMA-2 model (7b, chat) (Touvron et al., 2023) with a random subset of the C4 dataset (Raffel et al., 2020). All experiments are conducted with n-gram watermark key sampling ($n = 5$). Additionally, we include the Soft watermark (Kirchenbauer et al., 2023) in our comparison, although it does not achieve step-wise distortion-free performance. Notably, when $\beta = 0$, the Beta-watermark becomes identical to the permute-reweight watermark (Hu et al., 2023).

**Machine Translation.** For the machine translation task, we utilize the WMT'14 English (En) to Romanian (Ro) dataset, comprising 1,999 examples in the test set. We employ the Multilingual Bart (MBart) model (Liu et al., 2020) along with its official tokenizer.

**Text Summarization.** For text summarization, we utilize the test set from the CNN-DM corpus (Hermann et al., 2015), which contains 11,490 examples. We employ the BART-large model, which has 400 million parameters, and the LLaMA-2 model with 7 billion parameters.

**Text Generation.** In text generation, we adhere to the experimental setup described in Kirchenbauer et al. (2023). We use a random subset of the C4 dataset for generation prompts. Our model selection includes the LLaMA-2, which has 7 billion parameters.

**Watermark Setup.** Our experiments primarily compare the beta-watermark with three other distortion-free watermarks: inversa-sampling, Gumbel-reparametrization, and permute-reweight. Additionally, we include the Soft watermark (Kirchenbauer et al., 2023) in our comparison. For beta-watermark, we explore various $\beta$ values from the set $\{0, 0.05, 0.1, 0.2, 0.3\}$. For the Soft watermark (Kirchenbauer et al., 2023), we investigate green list bias $\delta$ values from $\{0.5, 1.0, 1.5, 2.0\}$ with a fixed green list separator $\gamma = 0.5$. For n-gram key sampling, we consider the most recent 5 tokens as the texture key. For example, when generating $x_4$ in response to $(x_1, x_2, x_3)$, the texture key includes $(x_1, x_2, x_3)$, given only three tokens are available. Texture key history resets before generating each batch. For cipher generation, we use SHA-256 as the hash function and a 1024-bit random bitstrings as the secret key sk, the watermark key is given by $k = (\text{sk}, \boldsymbol{x}_{i-5,i-1})$. The permutation $\pi$ is sampled using hash($k$) as the random seed. We also compare beta-watermark with inverse-sampling watermark Kuditipudi et al. (2023) and permute-reweight watermark Hu et al. (2023); Wu et al. (2023), following the settings in their open-sourced code[1][2].

**Evaluation Metrics for Text Quality.** In this part, we detail the metrics used to evaluate text quality:

- **ROUGE Score.** For the summarization task, we employ the ROUGE score (Lin, 2004), which measures the overlap of n-grams between the generated summaries and the reference texts to evaluate how effectively the summary captures the essential content.

---

[1] https://github.com/jthickstun/watermark
[2] https://github.com/xiaoniu-578fa6bff964d005/UnbiasedWatermark

Table 4: Performance of different watermarks under one-time generation. For each prompt, only one response is generated.

| | Text Summarization | | | Machine Translation | |
|---|---|---|---|---|---|
| | BERT Score↑ | ROUGE-1↑ | Perplexity↓ | BERT Score↑ | BLEU ↑ |
| No Watermark | 0.3174±0.0885 | 0.3772±0.0962 | 6.4155±3.3009 | 0.2683±0.1967 | 10.8705±10.1914 |
| Beta-Reweight (\beta=0) | 0.3162±0.0871 | 0.3758±0.0961 | 6.3810±3.2753 | 0.2669±0.1966 | 10.6208±9.5880 |
| Beta-Reweight (\beta=0.05) | 0.3171±0.0877 | 0.3760±0.0952 | 6.3986±3.2142 | 0.2683±0.1907 | 10.6511±10.1191 |
| Beta-Reweight (\beta=0.1) | 0.3169±0.0873 | 0.3762±0.0965 | 6.4250±3.2944 | 0.2687±0.1962 | 10.9058±10.5317 |
| Beta-Reweight (\beta=0.2) | 0.3184±0.0883 | 0.3771±0.0966 | 6.3889±3.2144 | 0.2641±0.1947 | 10.9852±10.7563 |
| Beta-Reweight (\beta=0.3) | 0.3167±0.0869 | 0.3764±0.0954 | 6.3972±3.2855 | 0.2668±0.1907 | 10.7865±9.8656 |
| Inverse-sampling | 0.3182±0.0876 | 0.3772±0.0964 | 6.3377±3.1274 | 0.2894±0.1869 | 11.6892±10.5368 |
| Gumbel-reparametrization | 0.3171±0.0868 | 0.3763±0.0961 | 6.3538±3.2221 | 0.3065±0.1875 | 11.8670±10.6599 |
| Soft($\delta$=0.5) | 0.3152±0.0862 | 0.3746±0.0949 | 6.4894±3.2453 | 0.2541±0.1950 | 10.3546±9.7336 |
| Soft($\delta$=1.0) | 0.3125±0.0856 | 0.3724±0.0937 | 6.8647±3.4364 | 0.2241±0.1922 | 9.5412±9.0065 |
| Soft($\delta$=1.5) | 0.3067±0.0825 | 0.3673±0.0917 | 7.4633±3.5928 | 0.1876±0.1891 | 8.5556±8.5925 |
| Soft($\delta$=2.0) | 0.2996±0.0805 | 0.3605±0.0899 | 8.4847±4.1598 | 0.1380±0.1750 | 6.9994±6.7528 |

- **BLEU score.** For the machine translation task, we rely on the BLEU score (Papineni et al., 2002), emphasizing the lexical similarity between machine-generated translations and human reference translations.

- **BERTScore.** BERTScore Zhang et al. (2019) calculates the similarity between two sentences by summing the cosine similarities of their token embeddings. We utilize BERTScore-F1, BERTScore-Precision, and BERTScore-Recall for assessing both text summarization and machine translation tasks.

- **Perplexity.** Perplexity, a concept from information theory, measures how well a probability model or distribution predicts a sample. It is used to compare the performance of probability models, where a lower perplexity indicates a more predictive model. We apply perplexity to evaluate both text summarization and text generation tasks.

**Evaluation Metrics for Detecting Efficiency of Watermarks.** In this section, we present the metrics used to evaluate the detectability of watermarks:

- **Type I and II Errors.** We employ the true positive rate (TPR), false positive rate (FPR), true negative rate (TNR), and false negative rate (FNR) to assess watermark detection across a mix of watermarked and non-watermarked sentences. The FPR measures the Type I error, which occurs when the null hypothesis is incorrectly rejected when it is actually true. The FNR measures the Type II error, where there is a failure to reject a false null hypothesis.

# F    ADDITIONAL EXPERIMENTAL RESULTS

In this section, we introduce the additional experiments conducted in our paper.

**Weakly Distortion-Free.** The full results are presented in Table 4. This figure shows that compared to the model without watermarks, all weakly distortion-free watermarks exhibit no significant performance bias in text summarization and text generation tasks. However, for the Soft-watermark, a significant performance bias is observable as $\delta$ increases. Besides, we also include a comprehensive results for the combination of all PDA-rules and all three kinds of key sampling methods under text generation tasks. The results are presented in Table 5. We also don't observe the distribution bias under the $\Delta$ metrics.

**Strongly Distortion-Free.** The full results are displayed in Table 6, where we include all PDA-rule and key sampling method into comparison. From this table, it is evident that compared to the no watermark model, all weakly distortion-free watermarks demonstrate performance bias across all tasks. In contrast, the Beta-watermark exhibits less bias compared to other weakly distortion-free watermarks. Additionally, as $\beta$ increases, the distribution bias is further reduced, consistent with our theoretical analysis.

Table 5: Performance of different watermarks under one-time generation for text generation tasks. For each prompt, only one response is generated

| PDA-rule | Watermark key | bertscore.precision | bertscore.recall | bertscore.f1 | ppl | rouge1 | rouge2 | rougeL |
|---|---|---|---|---|---|---|---|---|
| $\beta$-reweight($\beta$=0) | fixed key set | 0.3062±0.0954 | 0.3279±0.1019 | 0.3170±0.0880 | 6.4090±3.2113 | 0.3764±0.0960 | 0.1324±0.0808 | 0.2377±0.0793 |
| | n-gram hashing | 0.3048±0.0949 | 0.3276±0.1010 | 0.3162±0.0871 | 6.3810±3.2753 | 0.3758±0.0961 | 0.1314±0.0798 | 0.2372±0.0785 |
| | position hashing | 0.3050±0.0951 | 0.3271±0.1010 | 0.3160±0.0874 | 6.4285±3.2815 | 0.3759±0.0952 | 0.1315±0.0798 | 0.2374±0.0791 |
| $\beta$-reweight($\beta$=0.05) | fixed key set | 0.3061±0.0953 | 0.3289±0.1026 | 0.3174±0.0884 | 6.3903±3.3533 | 0.3764±0.0964 | 0.1327±0.0806 | 0.2385±0.0801 |
| | n-gram hashing | 0.3058±0.0944 | 0.3286±0.1021 | 0.3171±0.0877 | 6.3986±3.2142 | 0.3760±0.0952 | 0.1320±0.0797 | 0.2375±0.0785 |
| | position hashing | 0.3058±0.0951 | 0.3283±0.1021 | 0.3170±0.0876 | 6.4043±3.3037 | 0.3763±0.0959 | 0.1326±0.0797 | 0.2385±0.0789 |
| $\beta$-reweight($\beta$=0.1) | fixed key set | 0.3055±0.0948 | 0.3279±0.1014 | 0.3166±0.0873 | 6.4143±3.3500 | 0.3765±0.0956 | 0.1324±0.0795 | 0.2380±0.0785 |
| | n-gram hashing | 0.3054±0.0950 | 0.3285±0.1015 | 0.3169±0.0873 | 6.4250±3.2944 | 0.3762±0.0965 | 0.1327±0.0801 | 0.2377±0.0785 |
| | position hashing | 0.3060±0.0954 | 0.3285±0.1008 | 0.3172±0.0875 | 6.4214±3.2642 | 0.3762±0.0952 | 0.1322±0.0785 | 0.2382±0.0780 |
| $\beta$-reweight($\beta$=0.2) | fixed key set | 0.3068±0.0952 | 0.3296±0.1020 | 0.3181±0.0878 | 6.4131±3.3820 | 0.3778±0.0960 | 0.1337±0.0806 | 0.2395±0.0799 |
| | n-gram hashing | 0.3068±0.0958 | 0.3302±0.1026 | 0.3184±0.0883 | 6.3889±3.2144 | 0.3771±0.0966 | 0.1334±0.0811 | 0.2392±0.0794 |
| | position hashing | 0.3057±0.0949 | 0.3283±0.1025 | 0.3169±0.0880 | 6.3685±3.2764 | 0.3765±0.0963 | 0.1323±0.0800 | 0.2383±0.0794 |
| $\beta$-reweight($\beta$=0.3) | fixed key set | 0.3053±0.0955 | 0.3280±0.1018 | 0.3166±0.0878 | 6.3878±3.1945 | 0.3763±0.0954 | 0.1319±0.0799 | 0.2376±0.0788 |
| | n-gram hashing | 0.3052±0.0949 | 0.3284±0.1006 | 0.3167±0.0869 | 6.3972±3.2855 | 0.3764±0.0954 | 0.1325±0.0799 | 0.2379±0.0784 |
| | position hashing | 0.3066±0.0952 | 0.3288±0.1018 | 0.3176±0.0876 | 6.3845±3.2077 | 0.3771±0.0963 | 0.1327±0.0798 | 0.2385±0.0787 |
| Gumbel-reparametrization | fixed key set | 0.3011±0.0953 | 0.3277±0.1016 | 0.3143±0.0875 | 6.6430±3.5498 | 0.3746±0.0959 | 0.1309±0.0797 | 0.2361±0.0793 |
| | n-gram hashing | 0.3060±0.0942 | 0.3284±0.1011 | 0.3171±0.0868 | 6.3538±3.2221 | 0.3763±0.0961 | 0.1321±0.0797 | 0.2376±0.0788 |
| | position hashing | 0.3047±0.0958 | 0.3267±0.1019 | 0.3156±0.0881 | 6.4877±3.4127 | 0.3755±0.0957 | 0.1317±0.0800 | 0.2380±0.0790 |
| Inverse-sampling | fixed key set | 0.3063±0.0942 | 0.3297±0.1014 | 0.3179±0.0870 | 6.1846±3.1150 | 0.3777±0.0960 | 0.1334±0.0802 | 0.2391±0.0793 |
| | n-gram hashing | 0.3064±0.0953 | 0.3302±0.1018 | 0.3182±0.0876 | 6.3377±3.1274 | 0.3772±0.0964 | 0.1328±0.0809 | 0.2390±0.0799 |
| | position hashing | 0.3075±0.0962 | 0.3326±0.1022 | 0.3199±0.0881 | 6.2007±3.0213 | 0.3796±0.0960 | 0.1344±0.0813 | 0.2404±0.0802 |
| No Watermark | NA | 0.3058±0.0959 | 0.3293±0.1026 | 0.3174±0.0885 | 6.4155±3.3009 | 0.3772±0.0962 | 0.1328±0.0806 | 0.2388±0.0799 |
| Soft($\delta$=0.5) | n-gram hashing | 0.3013±0.0941 | 0.3294±0.1005 | 0.3152±0.0862 | 6.4894±3.2453 | 0.3746±0.0949 | 0.1310±0.0781 | 0.2362±0.0776 |
| Soft($\delta$=1.0) | n-gram hashing | 0.2956±0.0928 | 0.3296±0.0999 | 0.3125±0.0856 | 6.8647±3.4364 | 0.3724±0.0937 | 0.1279±0.0769 | 0.2328±0.0764 |
| Soft($\delta$=1.5) | n-gram hashing | 0.2858±0.0906 | 0.3280±0.0968 | 0.3067±0.0825 | 7.4633±3.5928 | 0.3673±0.0917 | 0.1229±0.0731 | 0.2271±0.0724 |
| Soft($\delta$=2.0) | n-gram hashing | 0.2751±0.0879 | 0.3246±0.0953 | 0.2996±0.0805 | 8.4847±4.1598 | 0.3605±0.0899 | 0.1158±0.0698 | 0.2207±0.0695 |

| PDA-rules | Watermark key | $\Delta$ bertscore.precision | $\Delta$ bertscore.recall | $\Delta$ bertscore.f1 | $\Delta$ ppl | $\Delta$ rouge1 | $\Delta$ rouge2 | $\Delta$ rougeL |
|---|---|---|---|---|---|---|---|---|
| $\beta$-reweight($\beta$=0) | fixed key set | 0.0694±0.0564 | 0.0674±0.0577 | 0.0625±0.0520 | 2.7242±2.8964 | 0.0700±0.0549 | 0.0585±0.0517 | 0.0606±0.0519 |
| | n-gram hashing | 0.0700±0.0561 | 0.0672±0.0567 | 0.0626±0.0513 | 2.7165±2.9231 | 0.0703±0.0560 | 0.0582±0.0517 | 0.0605±0.0519 |
| | position hashing | 0.0701±0.0565 | 0.0679±0.0575 | 0.0630±0.0518 | 2.7533±2.9858 | 0.0698±0.0554 | 0.0584±0.0521 | 0.0611±0.0533 |
| $\beta$-reweight($\beta$=0.05) | fixed key set | 0.0701±0.0570 | 0.0678±0.0569 | 0.0630±0.0519 | 2.7436±3.0276 | 0.0709±0.0550 | 0.0588±0.0521 | 0.0617±0.0527 |
| | n-gram hashing | 0.0700±0.0567 | 0.0679±0.0573 | 0.0631±0.0519 | 2.7419±2.9226 | 0.0701±0.0554 | 0.0583±0.0517 | 0.0606±0.0522 |
| | position hashing | 0.0703±0.0566 | 0.0685±0.0577 | 0.0631±0.0521 | 2.7540±2.9807 | 0.0713±0.0560 | 0.0590±0.0524 | 0.0616±0.0522 |
| $\beta$-reweight($\beta$=0.1) | fixed key set | 0.0695±0.0566 | 0.0674±0.0573 | 0.0623±0.0520 | 2.7563±3.0299 | 0.0693±0.0557 | 0.0580±0.0520 | 0.0608±0.0526 |
| | n-gram hashing | 0.0696±0.0563 | 0.0676±0.0567 | 0.0626±0.0515 | 2.7640±2.8903 | 0.0701±0.0558 | 0.0579±0.0516 | 0.0605±0.0520 |
| | position hashing | 0.0703±0.0566 | 0.0676±0.0571 | 0.0630±0.0518 | 2.7559±2.9446 | 0.0698±0.0555 | 0.0583±0.0513 | 0.0610±0.0515 |
| $\beta$-reweight($\beta$=0.2) | fixed key set | 0.0695±0.0560 | 0.0673±0.0570 | 0.0625±0.0512 | 2.7507±3.0184 | 0.0706±0.0553 | 0.0589±0.0524 | 0.0610±0.0525 |
| | n-gram hashing | 0.0698±0.0566 | 0.0679±0.0571 | 0.0629±0.0517 | 2.7376±2.9355 | 0.0699±0.0558 | 0.0589±0.0525 | 0.0607±0.0518 |
| | position hashing | 0.0699±0.0563 | 0.0688±0.0587 | 0.0632±0.0526 | 2.7001±2.9368 | 0.0697±0.0563 | 0.0584±0.0529 | 0.0608±0.0532 |
| $\beta$-reweight($\beta$=0.3) | fixed key set | 0.0706±0.0568 | 0.0680±0.0575 | 0.0631±0.0520 | 2.7242±2.9031 | 0.0701±0.0562 | 0.0581±0.0519 | 0.0608±0.0528 |
| | n-gram hashing | 0.0705±0.0566 | 0.0679±0.0570 | 0.0633±0.0515 | 2.7466±2.9944 | 0.0701±0.0552 | 0.0585±0.0514 | 0.0609±0.0527 |
| | position hashing | 0.0696±0.0559 | 0.0673±0.0565 | 0.0622±0.0510 | 2.7271±2.9034 | 0.0693±0.0552 | 0.0576±0.0507 | 0.0602±0.0513 |
| Gumbel-reparametrization | fixed key set | 0.0700±0.0572 | 0.0679±0.0578 | 0.0629±0.0524 | 2.8303±3.0803 | 0.0706±0.0561 | 0.0579±0.0523 | 0.0616±0.0530 |
| | n-gram hashing | 0.0694±0.0561 | 0.0678±0.0574 | 0.0625±0.0517 | 2.7221±2.9595 | 0.0708±0.0555 | 0.0588±0.0520 | 0.0607±0.0524 |
| | position hashing | 0.0702±0.0573 | 0.0682±0.0585 | 0.0630±0.0530 | 2.7680±3.0449 | 0.0702±0.0563 | 0.0593±0.0529 | 0.0615±0.0539 |
| Inverse-sampling | fixed key set | 0.0692±0.0555 | 0.0661±0.0564 | 0.0618±0.0508 | 2.6649±2.8626 | 0.0695±0.0556 | 0.0580±0.0516 | 0.0608±0.0520 |
| | n-gram hashing | 0.0697±0.0565 | 0.0674±0.0567 | 0.0625±0.0516 | 2.7131±2.8903 | 0.0705±0.0557 | 0.0581±0.0521 | 0.0603±0.0523 |
| | position hashing | 0.0704±0.0559 | 0.0677±0.0579 | 0.0628±0.0517 | 2.6266±2.8591 | 0.0698±0.0559 | 0.0583±0.0519 | 0.0612±0.0526 |
| Baseline | NA | 0.0701±0.0560 | 0.0674±0.0570 | 0.0628±0.0513 | 2.7535±2.9630 | 0.0707±0.0558 | 0.0583±0.0522 | 0.0613±0.0527 |
| Soft($\delta$=0.5) | n-gram hashing | 0.0700±0.0569 | 0.0677±0.0576 | 0.0627±0.0519 | 2.7403±2.9348 | 0.0700±0.0553 | 0.0581±0.0507 | 0.0606±0.0521 |
| Soft($\delta$=1.0) | n-gram hashing | 0.0692±0.0558 | 0.0666±0.0562 | 0.0616±0.0505 | 2.8607±3.0746 | 0.0688±0.0543 | 0.0569±0.0501 | 0.0595±0.0511 |
| Soft($\delta$=1.5) | n-gram hashing | 0.0704±0.0564 | 0.0661±0.0557 | 0.0613±0.0508 | 3.0427±3.1473 | 0.0688±0.0550 | 0.0566±0.0505 | 0.0593±0.0516 |
| Soft($\delta$=2.0) | n-gram hashing | 0.0736±0.0587 | 0.0669±0.0560 | 0.0635±0.0517 | 3.6349±3.6255 | 0.0699±0.0552 | 0.0576±0.0509 | 0.0601±0.0517 |

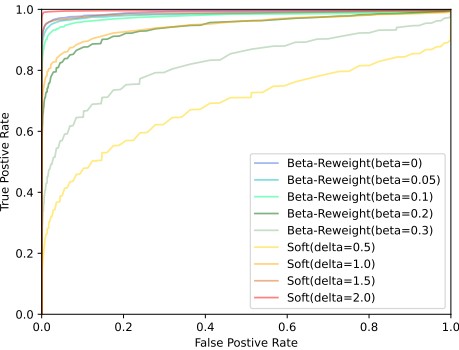

Figure 5: ROC curve of TPR vs FPR.

Table 6: Performance of different watermarks under multi-time generations. We randomly selected 1000 prompts and generated 100 responses for each. We use F1 scores of BERTScore and scale BERTScore and ROUGE-1 with a factor of 100.

| PDA-rule | Watermark key | Δ bertscore.precision | Δ bertscore.recall | Δ bertscore.f1 | Δ ppl | Δ rouge1 | Δ rouge2 | Δ rougeL |
|---|---|---|---|---|---|---|---|---|
| β-reweight(β=0) | fixed key set | 0.0070±0.0056 | 0.0066±0.0056 | 0.0062±0.0051 | 0.3123±0.2698 | 0.0071±0.0056 | 0.0062±0.0052 | 0.0062±0.0052 |
| | n-gram hashing | 0.0095±0.0082 | 0.0097±0.0084 | 0.0090±0.0077 | 0.3753±0.3448 | 0.0093±0.0078 | 0.0091±0.0087 | 0.0100±0.0093 |
| | position hashing | 0.0092±0.0077 | 0.0095±0.0085 | 0.0086±0.0077 | 0.3711±0.3339 | 0.0091±0.0075 | 0.0088±0.0086 | 0.0099±0.0093 |
| β-reweight(β=0.05) | fixed key set | 0.0074±0.0060 | 0.0070±0.0060 | 0.0066±0.0055 | 0.3084±0.2880 | 0.0073±0.0060 | 0.0061±0.0055 | 0.0063±0.0056 |
| | n-gram hashing | 0.0091±0.0074 | 0.0092±0.0076 | 0.0084±0.0071 | 0.3549±0.3200 | 0.0085±0.0070 | 0.0084±0.0079 | 0.0092±0.0082 |
| | position hashing | 0.0087±0.0070 | 0.0089±0.0078 | 0.0083±0.0068 | 0.3488±0.3192 | 0.0084±0.0067 | 0.0081±0.0073 | 0.0089±0.0083 |
| β-reweight(β=0.1) | fixed key set | 0.0066±0.0052 | 0.0066±0.0054 | 0.0060±0.0047 | 0.3061±0.2696 | 0.0069±0.0055 | 0.0059±0.0052 | 0.0061±0.0051 |
| | n-gram hashing | 0.0084±0.0071 | 0.0086±0.0070 | 0.0079±0.0065 | 0.3453±0.3214 | 0.0081±0.0068 | 0.0079±0.0073 | 0.0086±0.0078 |
| | position hashing | 0.0085±0.0069 | 0.0088±0.0073 | 0.0082±0.0066 | 0.3393±0.3195 | 0.0085±0.0066 | 0.0077±0.0069 | 0.0084±0.0074 |
| β-reweight(β=0.2) | fixed key set | 0.0072±0.0057 | 0.0069±0.0059 | 0.0065±0.0053 | 0.2960±0.2724 | 0.0073±0.0060 | 0.0062±0.0054 | 0.0062±0.0054 |
| | n-gram hashing | 0.0076±0.0060 | 0.0078±0.0063 | 0.0070±0.0057 | 0.3368±0.3231 | 0.0077±0.0061 | 0.0071±0.0064 | 0.0076±0.0066 |
| | position hashing | 0.0078±0.0064 | 0.0077±0.0063 | 0.0072±0.0059 | 0.3229±0.2906 | 0.0076±0.0062 | 0.0070±0.0065 | 0.0077±0.0067 |
| β-reweight(β=0.3) | fixed key set | 0.0066±0.0054 | 0.0066±0.0055 | 0.0060±0.0048 | 0.3078±0.2786 | 0.0069±0.0056 | 0.0059±0.0052 | 0.0060±0.0051 |
| | n-gram hashing | 0.0071±0.0056 | 0.0073±0.0058 | 0.0066±0.0052 | 0.3144±0.3015 | 0.0073±0.0060 | 0.0066±0.0056 | 0.0069±0.0058 |
| | position hashing | 0.0073±0.0059 | 0.0070±0.0060 | 0.0066±0.0054 | 0.3057±0.2991 | 0.0072±0.0059 | 0.0066±0.0057 | 0.0067±0.0058 |
| Gumbel-reparametrization | fixed key set | 0.0080±0.0063 | 0.0074±0.0059 | 0.0070±0.0047 | 0.3744±0.3205 | 0.0079±0.0064 | 0.0067±0.0060 | 0.0069±0.0057 |
| | n-gram hashing | 0.0480±0.0402 | 0.0461±0.0396 | 0.0428±0.0360 | 1.8892±1.8931 | 0.0488±0.0400 | 0.0409±0.0352 | 0.0427±0.0362 |
| | position hashing | 0.0494±0.0399 | 0.0485±0.0403 | 0.0442±0.0373 | 1.9935±2.4110 | 0.0512±0.0413 | 0.0423±0.0374 | 0.0442±0.0388 |
| Inverse-sampling | fixed key set | 0.0069±0.0054 | 0.0071±0.0061 | 0.0064±0.0054 | 0.3320±0.3066 | 0.0075±0.0061 | 0.0062±0.0057 | 0.0065±0.0052 |
| | n-gram hashing | 0.0486±0.0388 | 0.0481±0.0402 | 0.0439±0.0367 | 1.9380±2.0342 | 0.0499±0.0384 | 0.0403±0.0346 | 0.0428±0.0363 |
| | position hashing | 0.0503±0.0426 | 0.0469±0.0424 | 0.0448±0.0380 | 1.9095±2.2396 | 0.0491±0.0398 | 0.0422±0.0391 | 0.0441±0.0396 |
| Baseline | NA | 0.0068±0.0058 | 0.0067±0.0054 | 0.0062±0.0054 | 0.3028±0.2668 | 0.0070±0.0056 | 0.0060±0.0053 | 0.0060±0.0053 |
| Soft(δ=0.5) | n-gram hashing | 0.0078±0.0063 | 0.0069±0.0057 | 0.0064±0.0053 | 0.3331±0.2965 | 0.0076±0.0061 | 0.0065±0.0056 | 0.0065±0.0056 |
| Soft(δ=1.0) | n-gram hashing | 0.0127±0.0096 | 0.0086±0.0074 | 0.0091±0.0077 | 0.5473±0.4023 | 0.0099±0.0083 | 0.0090±0.0080 | 0.0090±0.0078 |
| Soft(δ=1.5) | n-gram hashing | 0.0200±0.0129 | 0.0106±0.0093 | 0.0128±0.0104 | 1.1237±0.5868 | 0.0136±0.0110 | 0.0123±0.0110 | 0.0127±0.0107 |
| Soft(δ=2.0) | n-gram hashing | 0.0312±0.0175 | 0.0133±0.0125 | 0.0195±0.0146 | 2.0817±0.8216 | 0.0194±0.0149 | 0.0182±0.0156 | 0.0188±0.0149 |

Table 7: AUC score of different watermarks under varying attack strength $\epsilon$ on text generation task.

| Beta-Reweight | $\epsilon$=0 | $\epsilon$=0.05 | $\epsilon$=0.1 | $\epsilon$=0.2 | $\epsilon$=0.3 |
|---|---|---|---|---|---|
| β=0 | 0.9948 | 0.9901 | 0.9742 | 0.8848 | 0.7447 |
| β=0.05 | 0.9912 | 0.9846 | 0.9672 | 0.8724 | 0.7312 |
| β=0.1 | 0.9889 | 0.9785 | 0.9550 | 0.8558 | 0.7078 |
| β=0.2 | 0.9796 | 0.9598 | 0.9201 | 0.7983 | 0.6735 |
| β=0.3 | 0.9447 | 0.9047 | 0.8509 | 0.7289 | 0.6191 |

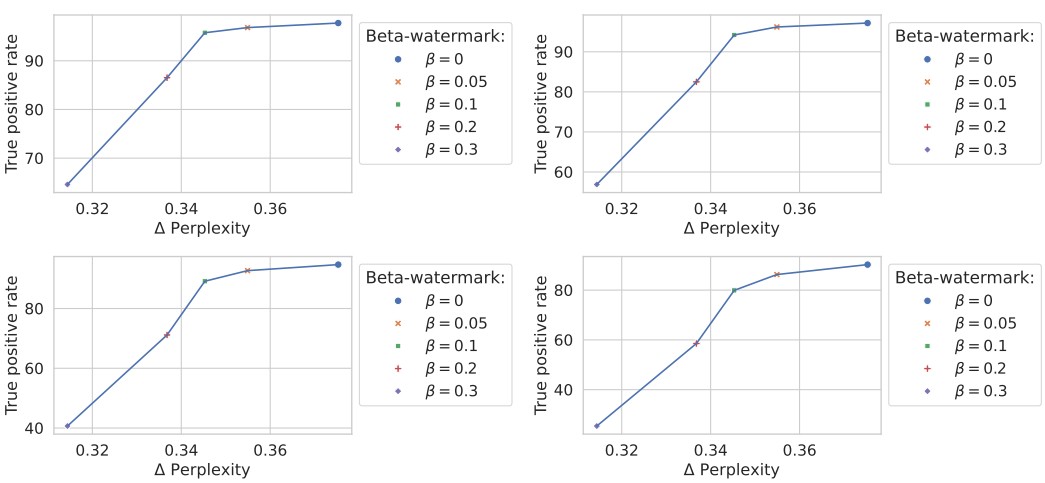

Figure 6: Trade-off between distribution bias and watermark strength under key collision. The TPR is measured under 10% (**Top Left**), 5% (**Top Right**), 1% (**Bottom Left**), 0.1% (**Bottom Right**) FPR. We can see Δ Perplexity (distribution bias) increase with the TPR.

Table 8: Empirical error rates for watermark detection on text generation. Each row is averaged over around 2000 watermarked examples.

|  |  | z=1.073 | | z=1.224 | | z=1.517 | | z=1.859 | |
|---|---|---|---|---|---|---|---|---|---|
|  |  | TNR↑ | TPR↑ | TNR↑ | TPR↑ | TNR↑ | TPR↑ | TNR↑ | TPR↑ |
| Soft-watermark | $\delta = 0.5$ | 90.00 | 46.05 | 95.00 | 38.78 | 99.00 | 24.41 | 99.90 | 13.04 |
|  | $\delta = 1$ | 90.00 | 88.37 | 95.00 | 85.02 | 99.00 | 76.80 | 99.90 | 68.42 |
|  | $\delta = 1.5$ | 90.00 | 97.15 | 95.00 | 96.65 | 99.00 | 94.64 | 99.90 | 90.90 |
|  | $\delta = 2$ | 90.00 | 99.45 | 95.00 | 99.39 | 99.00 | 99.06 | 99.90 | 97.90 |
| Beta-watermark | $\beta = 0$ | 90.00 | 97.75 | 95.00 | 97.17 | 99.00 | 94.69 | 99.90 | 90.25 |
|  | $\beta = 0.05$ | 90.00 | 96.82 | 95.00 | 96.19 | 99.00 | 92.67 | 99.90 | 86.26 |
|  | $\beta = 0.1$ | 90.00 | 95.76 | 95.00 | 94.19 | 99.00 | 89.13 | 99.90 | 79.90 |
|  | $\beta = 0.2$ | 90.00 | 86.53 | 95.00 | 82.49 | 99.00 | 71.14 | 99.90 | 58.55 |
|  | $\beta = 0.3$ | 90.00 | 64.59 | 95.00 | 56.88 | 99.00 | 40.67 | 99.90 | 25.38 |

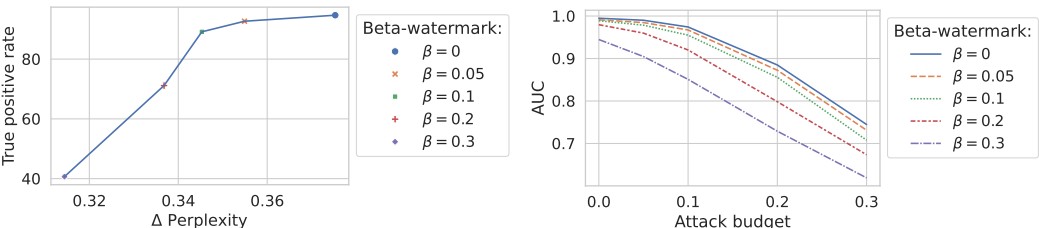

Figure 7: **Left.** Trade-off between distribution bias and watermark strength under key collision. The TPR is measured under 1% FPR. We can see $\Delta$ Perplexity (distribution bias) increase with the TPR. **Right.** AUC score of different watermarks under varying attack strength $\epsilon$ on text generation task.

### F.1 ABLATION STUDY

**Detect efficiency.** We compare the detection efficiency of beta-watermark with Soft-watermark on text generation tasks. We set the detecting scaling parameter (Definition 5.5) $C = 10$. We choose the threshold $z = 1.073, 1.224, 1.517, 1.859$, which corresponds to the 10%, 5%, 1% and 0.1% FPR. From Table 8, we see that the detect efficiency of beta watermark is comparable with the Soft-watermark (Kirchenbauer et al., 2023). We also see that when $\beta$ increases, the detection efficiency decreases, this is because a larger $\beta$ introduces a smaller distribution bias into the watermarked distribution, thus reducing the watermark strength.

We use the beta-watermark to illustrate the trade-off between watermark strength and distribution bias. As shown in Figure 6, with increasing values of $\beta$, the distribution bias decreases, but there is also a corresponding decrease in the true positive rate of watermark detection. This indicates that reducing the distribution bias of the watermark compromises its detectability.

In Figure 5, we see that the ROC of beta watermark is comparable with the Soft-watermark (Kirchenbauer et al., 2023). We also see that when $\beta$ increases, the detect efficiency decreases, this is because a larger $\beta$ introduces a smaller distribution bias into the watermarked distribution, thus reducing the watermark strength.

**Robustness.** We assessed the robustness of the beta-watermark against random text modifications and GPT-paraphrasing attacks (Kirchenbauer et al., 2023), where we modified 5%, 10%, 20%, and 30% (i.e., $\epsilon = 0.05, 0.1, 0.2, 0.3$) of the tokens. The results, as detailed in Figure F.1 (right), and Table 9, 10, 11 and 12 indicate that the beta-watermark maintains its robustness with moderate text modifications.

Table 9: Detectability and robustness of Soft watermark with beta-watermark under TPR @ FPR=0.1% on random token modification

| Random paraphrase | FPR=0.1% | $\epsilon$=0 | $\epsilon$=0.05 | $\epsilon$=0.1 | $\epsilon$=0.2 | $\epsilon$=0.3 |
|---|---|---|---|---|---|---|
| Beta-watermark | $\beta$=0 | 92.10 | 88.42 | 86.47 | 71.21 | 49.89 |
| | $\beta$=0.05 | 91.78 | 87.94 | 84.20 | 64.80 | 41.23 |
| | $\beta$=0.1 | 84.73 | 82.12 | 72.41 | 52.78 | 29.66 |
| | $\beta$=0.2 | 70.51 | 64.52 | 52.88 | 33.48 | 15.74 |
| | $\beta$=0.3 | 38.15 | 28.10 | 18.62 | 9.71 | 3.27 |
| Soft | $\delta$=0.5 | 13.59 | 9.45 | 5.41 | 2.53 | 1.38 |
| | $\delta$=1.0 | 69.32 | 61.03 | 51.62 | 33.14 | 15.68 |
| | $\delta$=1.5 | 92.52 | 88.78 | 84.47 | 69.16 | 44.33 |
| | $\delta$=2.0 | 98.35 | 97.58 | 96.37 | 90.65 | 73.60 |

Table 10: Detectability and robustness of Soft watermark with beta-watermark under TPR @ FPR=0.01% on random token modification

| Random paraphrase | FPR = 0.01% | $\epsilon$=0 | $\epsilon$=0.05 | $\epsilon$=0.1 | $\epsilon$=0.2 | $\epsilon$=0.3 |
|---|---|---|---|---|---|---|
| Beta-watermark | $\beta$=0 | 88.42 | 84.2 | 79.44 | 62.12 | 37.23 |
| | $\beta$=0.05 | 86.40 | 82.35 | 75.44 | 52.08 | 22.81 |
| | $\beta$=0.1 | 77.75 | 71.76 | 63.03 | 41.98 | 17.99 |
| | $\beta$=0.2 | 56.43 | 49.22 | 37.58 | 19.96 | 7.10 |
| | $\beta$=0.3 | 24.38 | 16.70 | 8.80 | 3.39 | 0.79 |
| Soft | $\delta$=0.5 | 6.80 | 4.03 | 1.96 | 0.81 | 0.35 |
| | $\delta$=1.0 | 57.78 | 50.06 | 38.63 | 19.71 | 6.27 |
| | $\delta$=1.5 | 88.21 | 83.56 | 78.00 | 56.92 | 29.59 |
| | $\delta$=2.0 | 97.36 | 95.82 | 93.07 | 84.82 | 63.36 |

Table 11: Detectability and robustness of Soft watermark with beta-watermark under TPR @ FPR=0.001% on random token modification

| Random paraphrase | FPR = 0.001% | $\epsilon$=0 | $\epsilon$=0.05 | $\epsilon$=0.1 | $\epsilon$=0.2 | $\epsilon$=0.3 |
|---|---|---|---|---|---|---|
| Beta-watermark | $\beta$=0 | 84.84 | 79.44 | 72.40 | 50.65 | 27.71 |
| | $\beta$=0.05 | 81.91 | 75.00 | 67.43 | 43.31 | 17.65 |
| | $\beta$=0.1 | 71.21 | 63.79 | 55.29 | 29.99 | 12.10 |
| | $\beta$=0.2 | 47.12 | 40.13 | 29.60 | 11.86 | 2.88 |
| | $\beta$=0.3 | 12.98 | 9.14 | 4.85 | 1.92 | 0.79 |
| Soft | $\delta$=0.5 | 3.23 | 2.30 | 1.27 | 0.35 | 0.35 |
| | $\delta$=1.0 | 47.82 | 37.96 | 29.11 | 10.86 | 2.80 |
| | $\delta$=1.5 | 82.77 | 77.66 | 71.09 | 47.51 | 19.05 |
| | $\delta$=2.0 | 96.15 | 93.62 | 90.54 | 78.22 | 50.83 |

Table 12: Detectability and robustness of Soft watermark with beta-watermark under TPR @ FPR=0.1% on GPT-4 paraphrase attack

| GPT-4 paraphrase | FPR=0.1% | $\epsilon$=0 | $\epsilon$=0.05 | $\epsilon$=0.1 | $\epsilon$=0.2 | $\epsilon$=0.3 |
|---|---|---|---|---|---|---|
| Beta-reweight | $\beta$=0 | 92.10 | 90.62 | 92.86 | 88.24 | 70.59 |
| | $\beta$=0.05 | 91.78 | 89.34 | 91.55 | 85.78 | 75.65 |
| | $\beta$=0.1 | 84.73 | 83.03 | 79.35 | 72.00 | 53.44 |
| | $\beta$=0.2 | 70.51 | 65.21 | 62.75 | 54.98 | 48.48 |
| | $\beta$=0.3 | 38.15 | 32.14 | 31.32 | 29.47 | 17.35 |
| Soft | $\delta$=0.5 | 13.59 | 11.98 | 11.69 | 13.08 | 7.41 |
| | $\delta$=1.0 | 69.32 | 65.83 | 64.37 | 61.36 | 55.12 |
| | $\delta$=1.5 | 92.52 | 91.14 | 92.46 | 89.38 | 76.70 |
| | $\delta$=2.0 | 98.35 | 97.79 | 98.58 | 95.93 | 95.00 |

