# OpenReview forum: "Pseudo- vs. True-Randomness: Rethinking Distortion-Free Watermarks of Language Models under Watermark Key Collisions"
_ICLR.cc/2025/Conference — ICLR 2025 Conference Withdrawn Submission_

### Official Review · Reviewer_SdCD · 2024-10-31

**Soundness:** 2
**Presentation:** 3
**Contribution:** 2
**Rating:** 5
**Confidence:** 4

**Summary:**

This paper explores the application of pseudo-random and true-random sampling in distortion-free watermarking for language models, highlighting that pseudo-random sampling may introduce distributional bias under watermark key collisions, thereby affecting watermark quality. The paper proposes a beta-watermark method, which balances watermark strength and distributional bias to mitigate collision-induced bias, enhancing detection efficiency.

**Strengths:**

1. This paper investigates the impact of pseudo-random versus true random sampling on watermarking applications, offering a valuable contribution to the field.
2. The paper provides a rigorous proof that strongly distortion-free watermarking algorithms are theoretically unattainable. Theoretical proofs are well-aligned with experimental results, supporting the paper’s main conclusions.
3. The proposed trade-off method between watermark strength and model performance demonstrates practical flexibility and applicability in specific scenarios.

**Weaknesses:**

1. The proposed beta-watermark shares considerable similarities with DiPmark (Yihan Wu, Zhengmian Hu, Hongyang Zhang, and Heng Huang. Dipmark: A Stealthy, Efficient and Resilient Watermark for Large Language Models), as both are variations of the permute reweight PDA-rule. Additionally, the algorithms in Appendix A are nearly identical to Algorithms 1 and 2 in DiPmark. It is recommended to add a comparison with DiPmark in the paper.
2. The experiments primarily compare against soft watermarking (ICML 2023 Best Paper) but lack examples of comparing watermark text effects under different parameters. Furthermore, it is recommended to include comparisons with other watermarking techniques mentioned in the paper to enhance understanding for the audience.
3. The paper does not analyze or compare watermark detection time, algorithmic complexity, or system performance overhead.
4. The paper analyzes the performance of the watermark model under key collisions, which reflects the issue of information capacity in watermarking algorithms. However, typical applications of text watermarking have low demands on key space, and the paper lacks a clear positioning of its problem setting. The paper uses identical prompts producing identical outputs to illustrate randomness issues, but this randomness is model parameter-dependent and unrelated to the watermarking algorithm.
5. The differences between Figures 4 and 6 are minimal. It is recommended to combine the different line plots into a single figure for comparison.

**Questions:**

1. Section 4.3 of the paper designs a black-box, distortion-free watermark detection algorithm that relies on comparisons with the original unwatermarked model. However, no comparative experiments are provided in the paper. What advantages does the proposed detection algorithm offer, and would it provide any efficiency improvements?
2. The experimental data only presents results with δ values greater than 1, in comparison to the reference Soft watermarking paper. However, no data is provided for δ < 1. Would the proposed method still demonstrate an advantage under δ < 1 conditions? Please provide relevant experimental data.
3. The discussion on pseudo-random versus true random sampling could be further expanded, with a more in-depth analysis on the specific impact of different pseudo-random algorithms on distribution bias.

---

> ### Author Response · Authors · 2024-11-16
> **Authors Response to Reviewer SdCD**
>
> > The proposed beta-watermark shares considerable similarities with DiPmark (Yihan Wu, Zhengmian Hu, Hongyang Zhang, and Heng Huang. Dipmark: A Stealthy, Efficient and Resilient Watermark for Large Language Models), as both are variations of the permute reweight PDA-rule. Additionally, the algorithms in Appendix A are nearly identical to Algorithms 1 and 2 in DiPmark. It is recommended to add a comparison with DiPmark in the paper.
>
> In Dipmark, the token probability interval $[0, 1]$ is divided into three regions: $[0, \alpha]$, $[\alpha, 1-\alpha]$, and $[1-\alpha, 1]$, with the probability mass within $[0, \alpha]$ being set to 0. As a result, compared to the beta watermark, Dipmark sacrifices the true randomness for tokens in the $[0, \alpha]$ range. We will include the relevant discussion and experimental results in our paper.
>
>
> > The experiments primarily compare against soft watermarking (ICML 2023 Best Paper) but lack examples of comparing watermark text effects under different parameters. Furthermore, it is recommended to include comparisons with other watermarking techniques mentioned in the paper to enhance understanding for the audience.
>
>
> Thank you for the suggestion. In Section F.1, we compare the effects of watermarking on text under different parameters for both the Soft watermark and the Beta watermark. Additionally, we have included comparisons with other watermarking techniques discussed in our paper. Please refer to Table 2, Table 5 (Appendix), and Table 6 (Appendix), where we compare the Beta watermark with the inverse-sampling, Gumbel-reparameterization, and permute-reweight watermark techniques.
>
>
> > The paper does not analyze or compare watermark detection time, algorithmic complexity, or system performance overhead.
>
>
> Following Dipmark, we show the watermark detection time on 1/1000 examples below. From the result, we see beta-watermark has similar detection time as soft watermark and dipmark.
> |  Number of samples | 1 | 1000 |
> |--------------------------|---------------------|-------|
> | Soft (Kirchenbauer et al)| 0.3s  | 92s |
> | Gumbel reparameterization (Kuditipudi et al)  | 80s | 12h |
> | Inverse-sampling / Permute-reweight (Hu et al)| 3.4s  | 412s |
> | DiPmark (Wu et al) | 0.3s |90s |
> | Beta-watermark     | 0.3s |90s|
>
>
>
> > The paper analyzes the performance of the watermark model under key collisions, which reflects the issue of information capacity in watermarking algorithms. However, typical applications of text watermarking have low demands on key space, and the paper lacks a clear positioning of its problem setting. The paper uses identical prompts producing identical outputs to illustrate randomness issues, but this randomness is model parameter-dependent and unrelated to the watermarking algorithm.
>
> Could you please elaborate further on the statement that “typical applications of text watermarking have low demands on key space”? In lines [165–178], we provide a detailed discussion of existing key sampling methods and key space considerations, and we demonstrate that key collision is inevitable under the current watermarking schemes.
>
> Moreover, the issue of “producing identical outputs” is not typically a problem with the model itself but rather with the pseudo-random sampling mechanism, which is directly tied to the watermarking algorithm. In the watermark generator, a pseudo-random sampling algorithm seeded by the watermark key is used to modify the token distribution. When the same seed (i.e., the same watermark key) is used, the pseudo-random sampling algorithm produces identical outputs. Therefore, the identical outputs are a direct result of the pseudo-random nature of the watermark generator combined with key collisions.
>
>
> > The differences between Figures 4 and 6 are minimal. It is recommended to combine the different line plots into a single figure for comparison.
>
> Thanks for the suggestion, we will merge them into one plot.
>
> > Section 4.3 of the paper designs a black-box, distortion-free watermark detection algorithm that relies on comparisons with the original unwatermarked model...
>
> To our best knowledge, this is the first algorithm that can detect whether an LM is watermarked or not given the black-box LM access, thus, we are not able to conduct comparison experiments. This algorithm enables the users to detect whether an LM is watermarked or not.
>
> > The experimental data only presents results with δ values greater than 1...
>
> Please check table 4,5,6 in the appendix, where we provide the result of soft watermark with $\delta=0.5$. From these tables we see that our method can still outperform soft watermark under $\delta<1$ conditions.
>
>
> > The discussion on pseudo-random versus true random sampling could be further expanded...
>
> In Theorem 4.9, we provide a detailed theoretical analysis about the distribution bias introduced by the existing three pseudo-random sampling algorithms. We would be happy to add more discussions based on the reviewer’s suggestions.

---

> > ### Author Response · Authors · 2024-11-20
> > **Comparison with DiPmark**
> >
> > This table illustrates the performance of beta watermarks and dipmark under multi-time generations, the experimental settings follow Table 2. From this table, we see that beta-watermarks introduce less distribution bias comparing to Dipmark.
> >
> > |                             | $\Delta$ Bert Score | $\Delta$ ROUGE-1 | $\Delta$ Perplexity |
> > |-----------------------------|:-------------------:|:----------------:|:-------------------:|
> > | Baseline                    |        0.0062       |      0.0070      |        0.3028       |
> > | Beta-Reweight ($\beta=0.1$) |        0.0079       |      0.0081      |        0.3453       |
> > | Beta-Reweight ($\beta=0.2$) |        0.0070       |      0.0077      |        0.3368       |
> > | Beta-Reweight ($\beta=0.3$) |        0.0066       |      0.0073      |        0.3144       |
> > | DiPmark ($\alpha=0.4$)      |        0.0083       |      0.0090      |        0.3489       |
> > | DiPmark ($\alpha=0.3$)      |        0.0078       |      0.0083      |        0.3452       |
> > | DiPmark ($\alpha=0.2$)      |        0.0070       |      0.0074      |        0.3225       |

---

> > ### Comment · Reviewer_SdCD · 2024-11-22
> >
> > 1. Thanks the authors for the detailed explanation of the connection with DiPmark in their response and for their plan to include relevant discussions and experimental results in the paper. However, the proposed algorithm in this paper shows a high degree of similarity to DiPmark in design. To ensure academic rigor and proper citation practices, I suggest that the authors explicitly cite previously published work (e.g., https://icml.cc/virtual/2024/poster/33605) in the algorithm description. Additionally, the authors should further analyze and clarify the specific differences between their algorithm and DiPmark, detailing the improvements and innovations proposed in this work. This will not only highlight the academic contributions of the paper but also provide readers with a clearer contextual understanding of the uniqueness of the proposed method.
> >
> > 2. I recommend including additional results showcasing the watermarking outcomes under different parameter settings, similar to Appendix H in Dipmark. Such results would facilitate a direct comparison with Dipmark, providing a clearer perspective on the relative performance and characteristics of the proposed method.
> >
> > 3. The original paper did not include the implementation mentioned above. I appreciate the authors for adding comparative experiments in their response.
> >
> > 4. (1) The statement " typical applications of text watermarking have low demands on key space " primarily refers to the practical focus of text watermarking schemes, which is often on distinguishing watermarked text from non-watermarked text or identifying specific sources, rather than pursuing an extremely large key space.
> > (2) It is recommended to include comparative experiments of generated outputs before and after embedding the watermark. This would provide a more intuitive demonstration of the watermarking algorithm's effects and its potential impact on generation quality.

---

> > > ### Author Response · Authors · 2024-11-23
> > > **Follow-up discussion**
> > >
> > > > Thanks the authors for the detailed explanation of the connection with DiPmark in their response and for their plan to include relevant discussions and experimental results in the paper. However, the proposed algorithm in this paper shows a high degree of similarity to DiPmark in design. To ensure academic rigor and proper citation practices, I suggest that the authors explicitly cite previously published work (e.g., https://icml.cc/virtual/2024/poster/33605) in the algorithm description. Additionally, the authors should further analyze and clarify the specific differences between their algorithm and DiPmark, detailing the improvements and innovations proposed in this work. This will not only highlight the academic contributions of the paper but also provide readers with a clearer contextual understanding of the uniqueness of the proposed method.
> > >
> > > >I recommend including additional results showcasing the watermarking outcomes under different parameter settings, similar to Appendix H in Dipmark. Such results would facilitate a direct comparison with Dipmark, providing a clearer perspective on the relative performance and characteristics of the proposed method.
> > >
> > > >The original paper did not include the implementation mentioned above. I appreciate the authors for adding comparative experiments in their response.
> > >
> > > Thank you for the suggestion, we will add suitable citation and discussion about DiPmark in our revision. We will also include the watermarked examples of beta watermark and dipmark in our paper.
> > >
> > > > (1) The statement " typical applications of text watermarking have low demands on key space " primarily refers to the practical focus of text watermarking schemes, which is often on distinguishing watermarked text from non-watermarked text or identifying specific sources, rather than pursuing an extremely large key space.
> > >
> > > We agree that one of the practical considerations for text watermarking schemes is on distinguishing watermarked text from non-watermarked text. However, another critical focus is preserving the quality of the watermarked content. A small key space increases the likelihood of key collisions, which can significantly degrade output quality. If a watermarking method compromises content quality too much, its practicality is diminished, making it unsuitable for industry-level applications.
> > >
> > > > (2) It is recommended to include comparative experiments of generated outputs before and after embedding the watermark. This would provide a more intuitive demonstration of the watermarking algorithm's effects and its potential impact on generation quality.
> > >
> > > We do have comparative experiments evaluating the generated outputs before and after embedding the watermark. Please refer to the "no watermark" results in Figure 3, the "Baseline" in Table 2 and Tables 4–6 in the appendix. Our findings show that for one-time generation, weakly distortion-free watermarks maintain the same generation quality as the non-watermarked LM. However, under single-prompt multi-time generation, we observe performance biases between the non-watermarked LM and the watermarked LMs.

---

> > > > ### Comment · Reviewer_SdCD · 2024-11-25
> > > >
> > > > Thanks Authors for the 2nd round of anwers!
> > > > Based upon the exsing materials in hand and all the feedbacks, there is no strong evidence to change the current rating.

---

### Official Review · Reviewer_WuLE · 2024-11-03

**Soundness:** 3
**Presentation:** 2
**Contribution:** 2
**Rating:** 3
**Confidence:** 4

**Summary:**

The authors propose three levels of distortion-freeness in watermarks, step-wise, weakly, and strongly distortion-freeness. They also argue that key collisions are inevitable in watermark generation, and then proceed to show that under key collisions, existing watermarks are not strongly distortion-free and confirm it as an impossibility result in theory. They also propose a metric for detecting watermarked models. Finally, they propose a new family of watermarks, and show that it is weakly distortion-free and performs better than existing ones on benchmark datasets.

**Strengths:**

1. The problem studied in this paper is important and it is also a timely topic. As the authors have argued, key collisions are inevitable, then how to resolve the problem brought by key collisions in watermarks is an important problem to study.
2. Exploiting true randomness within the usage of pseudorandom keys is a good insight for designing watermarks.

**Weaknesses:**

This paper is composed of two parts, proofs and claims on distortion-free watermarks and a new solution for weakly distortion-free watermarks. Both parts have limited contributions. See below for more details.
1. The technical merit is limited. The claims in this paper are rather easy to prove (yet the notation is quite heavy). E.g., Theorem 4.13 is pretty straightforward. A model with detectable watermarks and the same output distribution as the original one does not exist.

2. I do not understand the motivation for detecting watermarked models. Why not detect watermarked texts? The service provider knows which models (they provide) are watermarked in the first place, and the task is to determine which texts are generated from the watermarked models they provided (for copyright, intellectual property, plagiarism detection, etc.). Hence, theorem 4.12 (although it is technically valid) also does not make sense to me. Why would the service provider compute the performance metric for two models that she/he has access to and determine if one of them is watermarked?

3. The proposed solution ($0<\beta<0.5$) is similar to the permute-reweight baseline ($beta=0$). The novelty is limited and the performance increase (shown in Table 2) is marginal (compared with $beta=0$).

4. The experiment section is limited. The baselines are limited to distortion-free baselines and statistical watermarks are not included.

5. It is important to show that distortion does lead to worse performance in watermarked models, otherwise distortion-freeness is not well-motivated. Without this motivation, this paper is mainly proposing a new solution for distortion-free watermarks, and given that the solution does not seem novel, the contribution of this paper is limited.

6. From Figure 3, it is not clear which watermark performs better. Is there any direct measurement of the distortion-freeness?

**Questions:**

1. Address the above weakness section, especially W4, W5, W6, W2.
2. There are some typos. E.g., in line 60, ``In particular`` should not be capitalized.

---

> ### Author Response · Authors · 2024-11-16
> **Response to Reviewer WuLE**
>
> > The technical merit is limited. The claims in this paper are rather easy to prove (yet the notation is quite heavy). E.g., Theorem 4.13 is pretty straightforward. A model with detectable watermarks and the same output distribution as the original one does not exist.
>
> We respectfully disagree. The technical merit of a theorem should not be judged based on the simplicity of its proof but rather on the significance of the insights it provides. To the best of our knowledge, we are the first to theoretically prove the non-existence of a strongly distortion-free watermark, which is a critical finding for the LLM watermarking community. Furthermore, we would appreciate it if the reviewer could elaborate on why they find Theorem 4.13 to be straightforward.
>
>
> > I do not understand the motivation for detecting watermarked models. Why not detect watermarked texts? The service provider knows which models (they provide) are watermarked in the first place, and the task is to determine which texts are generated from the watermarked models they provided (for copyright, intellectual property, plagiarism detection, etc.). Hence, theorem 4.12 (although it is technically valid) also does not make sense to me. Why would the service provider compute the performance metric for two models that she/he has access to and determine if one of them is watermarked?
>
> We apologize for any confusion. The intuition behind detecting watermarked models is to enable **users** to determine whether a language model is watermarked or not. For instance, consider a scenario where a user has access to the original LM provided by an official service (e.g., LLAMA or ChatGPT) and the same model offered by a private institution at a lower cost. Our method allows the user to detect whether the model from the private institution has been watermarked or modified.
>
>
>
>
> > The proposed solution (0<β<0.5) is similar to the permute-reweight baseline (beta=0). The novelty is limited and the performance increase (shown in Table 2) is marginal (compared with beta=0).
>
> We respectfully disagree with this comment. The beta watermark is built upon the permute-reweight watermark; however, the performance improvement is substantial compared to $\beta=0$. For instance, in table 2, when comparing $\beta=0.3$ to $\beta=0$, $\beta=0.3$ achieves at least a 30% reduction in the $\Delta$ BERT Score, which is clearly a significant improvement and far from marginal.
>
>
> > The experiment section is limited. The baselines are limited to distortion-free baselines and statistical watermarks are not included.
>
> As this paper focuses on distortion-free watermarks, we believe we have included all relevant distortion-free watermarks in our comparison. Since other statistical watermarks are not even step-wise distortion-free, they are not pertinent to the distortion-freeness experiments conducted in our work.
>
> > It is important to show that distortion does lead to worse performance in watermarked models, otherwise distortion-freeness is not well-motivated. Without this motivation, this paper is mainly proposing a new solution for distortion-free watermarks, and given that the solution does not seem novel, the contribution of this paper is limited.
>
> Distortion does lead to worse performance in watermark models, which has already been demonstrated by [1,2]. In Figure 3 and Table 4. We used the same settings in [1,2] and showed that non-distortion-free watermarks can reduce the performance of LMs on machine translation and text summarization tasks.
>
>
> > From Figure 3, it is not clear which watermark performs better. Is there any direct measurement of the distortion-freeness?
>
> We apologize for the unclear figure. We have also summarized the performance comparison in appendix Table 4, where we observe a significant performance drop for non-distortion-free watermarks. Currently, there are no direct metrics available to measure distortion-freeness. Following the methodology of existing works [1,2], we evaluate distortion-freeness based on the quality of the generated content.
>
>
>
>
> >questions.
>
>
> Thank you for pointing out the typos, we will correct them in our revision.
>
>
> [1] Hu et al. Unbiased watermark for large language models.
> [2] Wu et al. Dipmark: a stealthy, efficient and resilient watermark for large language models.

---

> > ### Comment · Reviewer_WuLE · 2024-12-01
> >
> > Thanks for the response.
> >
> > The response does not address my questions directly. In particular, I am not convinced that the proposed solution is novel (it is the permute-reweight watermark with an additional parameter), and the technical part (lemmas and proofs) does not seem interesting enough to clear the bar of ICLR. I will keep my score.

---

### Official Review · Reviewer_DSq6 · 2024-11-04

**Soundness:** 3
**Presentation:** 3
**Contribution:** 2
**Rating:** 5
**Confidence:** 3

**Summary:**

LLM watermarking use pseudorandom sampling to embed statistical signals into generated content without distorting quality. However, key collisions in pseudorandom sampling can lead to correlations that undermine this distortion-free property. The author show that key collisions are unavoidable and cause existing watermarks to bias towards the original LLM distribution. They prove that a perfect distortion-free watermark is unattainable and introduce beta-watermarks, which effectively reduce distribution bias under key collisions.

**Strengths:**

The paper is well written and the contributions are clear. It is nice to have a paper that clearly states what "distortion-free" can mean for LLM watermarks,  what are the implications etc. The math proofs looked correct to me.

**Weaknesses:**

The paper concludes that creating a fully distortion-free watermark is impossible because of key collisions. This result appears aligns with expectations, and it may not hold significant interest for the LLM watermarking community.


In particular, the importance of strongly distortion-free watermarks remains unclear. The authors suggest that a non-distortion-free watermark would result in repetitive generations. However, this issue also arises without watermarking when using greedy decoding. Furthermore, Aaranson et al. employ random decoding on top of the modified distribution of the next token, not just greedy decoding, which is the sole focus of the discussions regarding Aaranson et al. In such cases, repeatedly prompting a model with the same input does not necessarily produce identical outputs. Is this understanding correct?

**Questions:**

See weaknesses.
 - "Our studies reveal that key collisions are inevitable due to the limited availability of watermark keys, and existing distortion-free watermarks exhibit a significant distribution bias toward the original LM distribution in the presence of key collisions." -> What do the authors mean by "towards the original LM distribution?"

---

> ### Author Response · Authors · 2024-11-16
> **Response to Reviewer DSq6**
>
> > The paper concludes that creating a fully distortion-free watermark is impossible because of key collisions. This result appears aligns with expectations,
>
> Thank you for recognizing the correctness of our result!
>
>
> > and it may not hold significant interest for the LLM watermarking community.
>
> In our work, we introduced three levels of distortion-freeness and re-evaluated the existing distortion-free watermark. We demonstrate that all of these approaches are only weakly distortion-free, rather than strongly distortion-free, indicating that they still introduce distributional bias into the language model. We believe this insight is crucial for advancing the understanding of distortion-freeness in LM watermarking.
>
>
> > In particular, the importance of strongly distortion-free watermarks remains unclear. The authors suggest that a non-distortion-free watermark would result in repetitive generations. However, this issue also arises without watermarking when using greedy decoding.
>
> Strongly distortion-free watermarks are crucial because they preserve the quality of content generated by language models, making them more suitable for industry-level LLM applications where watermarking is necessary. Moreover, "repetitive generations" are merely one consequence of non-distortion-free watermarks. The core issue with non-distortion-free watermarks lies in their intrinsic bias, which distorts the original LM distribution and ultimately degrades generation quality.
>
>
>
>
> > Furthermore, Aaranson et al. employ random decoding on top of the modified distribution of the next token, not just greedy decoding, which is the sole focus of the discussions regarding Aaranson et al. In such cases, repeatedly prompting a model with the same input does not necessarily produce identical outputs. Is this understanding correct?
>
> No, in Theorem 4.13, we theoretically prove the non-existence of strongly distortion-free watermarks beyond the key collision assumption. This implies that the approach by Aaronson et al. is also not strongly distortion-free and introduces distributional bias into language models.
>
>
> > "Our studies reveal that key collisions are inevitable due to the limited availability of watermark keys, and existing distortion-free watermarks exhibit a significant distribution bias toward the original LM distribution in the presence of key collisions." -> What do the authors mean by "towards the original LM distribution?"
>
> Because the watermarking method will change the sampling process of the LM, the distribution of watermarked LMs will be different from the distribution of non-watermarked LMs.

---

> > ### Comment · Reviewer_DSq6 · 2024-11-19
> > **Thank you**
> >
> > Thank you for your answer.
> >
> > I still don't get why prompting several times an LLM that watermarks its outputs will yield the same results over and over again.  Could the authors clarify that point?

---

> > > ### Author Response · Authors · 2024-11-19
> > > **Follow-up discussion**
> > >
> > > Thank you for actively engaging in the discussion! Below, we address your new concerns.
> > >
> > > > I still don't get why prompting several times an LLM that watermarks its outputs will yield the same results over and over again. Could the authors clarify that point?
> > >
> > > This occurs because watermark key collisions cause the pseudo-randomness of watermark sampling algorithms to become deterministic. Consider the inverse-sampling watermark as an example: in this method, the LM token probabilities are first mapped onto the interval [0,1] (see Figure 1, left). A pseudo-random number generator, seeded with the watermark key
> > > $k$, is then used to generate a number within this interval. The token corresponding to the location of the pseudo-random number within the its probability intervals in [0,1] is selected.
> > >
> > > However, when repeatedly prompting a watermarked LLM (using inverse sampling), the same watermark key is used to seed the pseudo-random number generator. This results in the generation of the same pseudo-random number and consequently the same tokens, leading to identical outputs across prompts.

---

> > > > ### Comment · Reviewer_DSq6 · 2024-11-23
> > > >
> > > > Thank you. But for Aaranson et al., the seed + watermark window determines a float r \in )0,1(, and then the next token can be sampled from r^{1/p} using any other seed which does not have anything to do with the watermarking scheme. In that case, prompting the model several times with the same prompts does not lead to the same answer, so I still don't get the author's point.

---

> > > > > ### Author Response · Authors · 2024-11-23
> > > > > **Follow-up discussion**
> > > > >
> > > > > > Thank you. But for Aaranson et al., the seed + watermark window determines a float r \in )0,1(, and then the next token can be sampled from r^{1/p} using any other seed which does not have anything to do with the watermarking scheme. In that case, prompting the model several times with the same prompts does not lead to the same answer, so I still don't get the author's point.
> > > > >
> > > > > Thank you for your comments. Assuming the token set is $\{t_1,...,t_n\}$, in Aaronson et al, the seed + watermark window determines $n$ float numbers $r_i \in [0,1]$, where $n$ is the number of tokens. Then, we sample the next token $t_i$ which maximize $r_i^{1/p_i}$, where $p_i$ is the token probability of $t_i$ (see the 7-th slides of Aaronson et al). Thus, the next token is not sampled with another seed. With the watermark key collision, the watermark algorithm will keep generating the same float numbers $r_i$, and $\max_i r_i^{1/p_i}$ will produce the same value and the generated answers will be the same.

---

### Note · Authors · 2025-01-22

I have read and agree with the venue's withdrawal policy on behalf of myself and my co-authors.